Molecular evolution in the membrane

# NALCN/Cch1 channelosome subunits originated in early eukaryotes

Adriano Senatore[1,2] , Tatiana D. Mayorova[1] , Luis A. Yañez-Guerra[3,4] , Wassim Elkhatib[1,2] , Brian Bejoy[1,2] , Philippe Lory[5] , and Arnaud Monteil[5,6]

The sodium leak channel NALCN, a key regulator of neuronal excitability, associates with three ancillary subunits that are critical for its function: a subunit called FAM155, which interacts with the extracellular regions of the channel, and two cytoplasmic subunits called UNC79 and UNC80. Interestingly, NALCN and FAM155 have orthologous phylogenetic relationships with the fungal calcium channel Cch1 and its subunit Mid1; however, UNC79 and UNC80 have not been reported outside of animals. In this study, we leveraged expanded gene sequence data available for eukaryotes to reexamine the evolutionary origins of NALCN and Cch1 channel subunits. Our analysis corroborates the direct phylogenetic relationship between NALCN and Cch1 and identifies a larger clade of related channels in additional eukaryotic taxa. We also identify homologues of FAM155/Mid1 in Cryptista algae and UNC79 and UNC80 homologues in numerous non-metazoan eukaryotes, including basidiomycete and mucoromycete fungi and the microbial eukaryotic taxa Apusomonadida, Malawimonadida, and Discoba. Furthermore, we find that most major animal lineages, except ctenophores, possess a full complement of NALCN subunits. Comparing structural predictions with the solved structure of the human NALCN complex supports orthologous relationships between metazoan and non-metazoan FAM155/Mid1, UNC79, and UNC80 homologues. Together, our analyses reveal unexpected diversity and ancient eukaryotic origins of NALCN/Cch1 channelosome subunits and raise interesting questions about the functional nature of this channel complex within a broad, eukaryotic context.

## Introduction

The sodium leak channel NALCN represents a fourth major branch of four-domain cation channels in animals, the others being low-voltage activated calcium channels (i.e., Ca$_V$3 channels), high-voltage activated calcium channels (Ca$_V$1 and Ca$_V$2), and voltage-gated sodium channels (Na$_V$) (Liebeskind et al., 2012; Pozdnyakov et al., 2018). Studies in several animals, primarily nematode worms, fruit flies, and mice, have revealed critical functions for NALCN in processes including sleep, circadian rhythm, breathing, nociception, pain, locomotion, and parturition (Ren, 2011; Cochet-Bissuel et al., 2014; Monteil et al., 2024). At the cellular level, NALCN contributes depolarizing leak sodium currents that help set the resting membrane potential of neurons and other excitable cells, and its activity can be modulated to exert changes in cellular excitability (Monteil et al., 2024). Both *de novo* dominant and inherited recessive pathogenic variants of NALCN are described in severe pathological conditions in humans characterized by a wide range of symptoms, and NALCN is implicated in many other diseases, including psychiatric disorders and cancer (Monteil et al., 2024). *In vivo* and *in vitro*, NALCN does not operate on its own but as part of a large multi-protein complex herein referred to as the NALCN channelosome. Specifically, NALCN associates with a subunit called FAM155 and two large cytoplasmic subunits, UNC79 and UNC80 (Kang and Chen, 2022; Kschonsak et al., 2022; Zhou et al., 2022). These additional ancillary proteins are necessary for the functional expression of NALCN, its trafficking to the plasma membrane, its cellular localization, and indeed the stability of the entire channelosome complex (reviewed in Ren [2011]; Cochet-Bissuel et al. [2014]; Monteil et al. [2024]).

[1]Department of Biology, University of Toronto Mississauga, Mississauga, Canada; [2]Department of Cell and Systems Biology, University of Toronto, Toronto, Canada; [3]School of Biology, University of Southampton, Southampton, UK; [4]Institute for Life Sciences, University of Southampton, Southampton, UK; [5]Institut de Génomique Fonctionnelle, Université de Montpellier, CNRS, INSERM, Montpellier, France; [6]Department of Physiology, Faculty of Medicine Siriraj Hospital, Mahidol University, Bangkok, Thailand.

Correspondence to Adriano Senatore: adriano.senatore@utoronto.ca

This work is part of a special issue on Molecular Evolution in the Membrane: Ion Channels, Transporters, and Receptors.

Interestingly, previous analyses revealed that NALCN and FAM155 have phylogenetic orthologues in fungi, where they are referred to as Cch1 and Mid1, respectively (Liebeskind et al., 2012; Ghezzi et al., 2014; Pozdnyakov et al., 2018). In fungi, Cch1 and Mid1 form a highly regulated $Ca^{2+}$ permeable channel that restores calcium levels in response to various stimuli, including endoplasmic reticulum stress and pheromone signaling, a function that contributes to the virulence of several pathogenic fungi, including *Cryptococcus neoformans*, *Candida albicans*, and *Aspergillus fumigatus* (Peiter et al., 2005; Yu et al., 2012; Harren and Tudzynski, 2013; De Castro et al., 2014; Vu et al., 2015). Instead, UNC79 and UNC80, which form a heterodimer in the cytoplasm, have not been reported outside of animals. In this study, we took advantage of the expanded genome and transcriptome sequence data that are currently available to re-interrogate the presence of NALCN subunits in animals, fungi, and other eukaryotes. Our results reveal an unexpected diversity of ancillary NALCN/Cch1 channelosome subunits in eukaryotes, including complete conservation of a tetrameric complex in animals, fungi, and apusomonads.

## Materials and methods

### Sequence identification

We trained custom hidden Markov models (HMMs) to search for homologues of NALCN, FAM155/Mid1, UNC79, and UNC80 within a set of high-quality eukaryotic proteomes spanning Amorphea (e.g., animals, choanoflagellates, and fungi; 88 species) and Diaphoretickes (plants and stramenopiles; 96 species), and separately, the complete set of fungal proteomes available in the integrated and functional genomic database FungiDB (release 68 [Basenko et al., 2018]). For the eukaryotic proteomes, details about sources and BUSCO (Simão et al., 2015) quality metrics are provided in Table S1. Collections of metazoan and fungal sequences were used to train the HMMs, manually extracted from available sequence data of various animal and fungal species in the NCBI nonredundant (Sayers et al., 2021) and FungiDB (Basenko et al., 2018) databases via BLAST (Altschul et al., 1990). The extracted sequences were confirmed as orthologous to target NALCN/Cch1 subunits via reciprocal BLAST of the NCBI nonredundant database and SmartBLAST. Details about the general composition of these different sequence collections are provided in the results, and all sequences are available in FASTA format in Data S2 (metazoan and fungal NALCN/Cch1), Data S3 (metazoan and fungal FAM155/Mid1), Data S4 (metazoan and fungal UNC80), Data S5 (fungal UNC80 only), Data S6 (metazoan and fungal UNC79), and Data S7 (fungal UNC79 only). Subsequently, these sequence sets were used as inputs for training subunit-specific custom HMMs using HMMER version 3.3.2, and the respective sequences identified with each model within the eukaryotic and FungiDB data sets were combined into corresponding sets and processed with CD-HIT (Li and Godzik, 2006) using a sequence identity threshold of 99.9% to remove redundant sequences. The NALCN/Cch1 sequences were further processed by predicting transmembrane helices with Phobius (Käll et al., 2004) and discarding those with <18 transmembrane helices to exclude single and tandem

domain pore-loop channels from downstream analyses, and a preliminary cluster analysis with CLuster ANalysis of Sequences (CLANS) to remove sequences with less than three sequence similarity connections with other sequences. All final sets of identified NALCN/Cch1, FAM155, UNC79, and UNC80 sequences are provided in Data S8 (NALCN/Cch1 homologues from eukaryotes), Data S9 (Cch1 homologues from fungi), Data S10 (FAM155/Mid1 homologues from eukaryotes), Data S11 (Mid1 homologues from fungi), Data S12 (UNC80 homologues from eukaryotes), Data S13 (UNC80 homologues from fungi), Data S14 (UNC79 homologues from eukaryotes), and Data S15 (UNC79 homologues from fungi).

For NALCN subunit searches strictly within animals, most protein sequences were identified through BLAST searches of the NCBI nonredundant database or the NCBI transcriptome shotgun assembly database using NALCN, FAM155, UNC79, and UNC80 protein sequences from humans, *Caenorhabditis elegans*, and *Trichoplax adhaerens* as queries. The exceptions are the identified *T. adhaerens* sequences, which were extracted from a whole animal mRNA transcriptome (Wong et al., 2019, *Preprint*); the *Ptychodera flava* (hemichordate) NALCN and FAM155 sequences, which were extracted from a gene model database derived from a genome sequencing effort (Simakov et al., 2015); and the UNC79 sequence from the sponge *Amphimedon queenslandica*, which was fragmented on NCBI and was hence extracted from an available transcriptome assembly (Fernandez-Valverde et al., 2015). Candidate sequences identified with BLAST were verified as orthologous through reciprocal BLAST of the NCBI nonredundant database as well as SmartBLAST. All verified sequences used for downstream analysis are provided in Data S16 (NALCN), Data S17 (FAM155), Data S18 (UNC80), and Data S19 (UNC79).

### Cluster analysis and phylogenetic inference

The various output files from our HMM searches through the eukaryotic proteome set (Data S8, S10, S12, and S14) were analyzed with the BLAST-based all-against-all clustering algorithm CLANS (Zimmermann et al., 2018), using the following amino acid substitution matrices and expect-value cutoffs: NALCN/Cch1 - PAM30, 1E-20; FAM155/Mid1 - BLOSUM45, 1E-6; UNC79 - BLOSUM65, 1E-5; UNC80 - BLOSUM65, 1E-5. The noted substitution matrices were chosen to reflect differences in sequence conservation among subunit homologues (i.e., PAM30 for the four-domain channels with their highly conserved transmembrane helices and BLOSUM45 and BLOSUM65 for FAM155 and UNC79/UNC80, respectively, due to their more divergent nature). Expect values were selected to ensure that all included sequences were linked to at least two other nodes. All cluster diagrams were annotated with the CLANS graphical user interface and exported as SVG files, and these were further annotated with Adobe Illustrator 2025 for generating figures.

For phylogenetic analyses, the various output files from our HMM and reciprocal BLAST searches (i.e., Data S8, S9, S10, S11, S12, S13, S14, S15, S16, S17, S18, and S19) were first aligned with the program MAFFT version 7.490 (Katoh and Standley, 2013), then trimmed with trimAl (Capella-Gutiérrez et al., 2009) using gappyout mode. These trimmed alignments were then

used as input for the maximum likelihood inference algorithm IQ-TREE2 (Minh et al., 2020), using the ModelFinder (Kalyaanamoorthy et al., 2017) option to identify the best-fit model for phylogenetic inference under the Bayesian information criterion. This resulted in the following best-fit models: NALCN/Cch1 in eukaryotes - Q.pfam+R10; Cch1 in fungi - Q.yeast+F+I+R8; NALCN in metazoans - Q.yeast+F+I+G4; FAM155/Mid1 in eukaryotes - VT+I+G4; Mid1 in fungi - Q.pfam+R8; FAM155 in metazoans - VT+R3; UNC80 in eukaryotes and fungi - Q.insect+F+I+R6; UNC80 in fungi - Q.yeast+F+I+G4; UNC80 in metazoans - Q.insect+F+R4; UNC79 in eukaryotes and fungi - Q.insect+R6; UNC79 in fungi - Q.insect+F+I+G4; UNC79 in metazoans - Q.insect+R4. For all trees, node support was estimated via 1000 replicate Shimodaira–Hasegawa approximate likelihood ratio tests (Guindon et al., 2010), approximate Bayes tests (Anisimova et al., 2011), and fast local bootstrap probability tests (Adachi and Hasegawa, 1996). All support values are listed as percentages. Resulting phylogenetic trees were annotated with the programs FigTree version 1.4.4 and the Interactive Tree Of Life version 7 (Letunic and Bork, 2007). Further annotation of the trees and final figure preparations were done using Adobe Illustrator 2025. Raw phylogenetic trees, in nexus format, are provided in Datas S20 (Fig. 2 A), S21 (Fig. 2 B), S22 (Fig. 3 B), S23 (Fig. 3 C), S24 (Fig. 5 B), S25 (Fig. 5 C), S26 (Fig. 6 B), S27 (Fig. 6 C), S28 (Fig. S1), S29 (Fig. S2), S30 (Fig. S3), and S31 (Fig. S4).

### Structure prediction and analysis

All structural predictions were done using AlphaFold3 (Abramson et al., 2024). Corresponding sequence accession numbers, along with predicted template modeling (pTM) and interface pTM scores, are provided in Table S2. Structures were analyzed and visualized with the program ChimeraX version 1.9 (Meng et al., 2023), and images for figures were exported as .png files and further annotated with Adobe Illustrator 2025. Multiple sequence alignments were done with Clustal Omega (Sievers et al., 2011) and annotated with Excel and Adobe Illustrator.

### Online supplemental material

Figs. S1, S2, S3 and S4 contain the maximum likelihood phylogenetic trees of NALCN, FAM155, UNC80, and UNC79 protein sequences from metazoans, respectively. Fig. S5 is a structural alignment of the solved human NALCN–FAM155 cryo-EM complex (PDB accession number 7sx4) and the same complex predicted with AlphaFold 3. Fig. S6 contains all the AlphaFold predicted structures generated in this study, colored according to their pLDDT confidence scores. Table S1 provides details about the eukaryotic proteomes used in this study. Table S2 provides details about species, accession numbers, and AlphaFold confidence scores for sets of proteins analyzed in this study. Data S1 provides protein sequences and a global alignment of the UNC80 homologues from human and the fungal species *R. delemar*. Data S2 provides protein sequences of metazoan and fungal NALCN/Cch1 subunits used to generate a profile HMM model. Data S3 provides protein sequences of metazoan and fungal FAM155/Mid1 subunits used to generate a profile HMM model. Data S4 provides protein sequences of metazoan and fungal UNC80

subunits used to generate a profile HMM model. Data S5 provides protein sequences of fungal UNC80 subunits used to generate a profile HMM model. Data S6 provides protein sequences of metazoan and fungal UNC79 subunits used to generate a profile HMM model. Data S7 provides protein sequences of fungal UNC79 subunits used to generate a profile HMM model. Data S8 provides protein sequences of identified four domain channels, including NALCN and Cch1, from the set of eukaryotic proteomes. Data S9 provides protein sequences of identified Cch1 homologues from the set of fungal proteomes. Data S10 provides protein sequences of identified FAM155/Mid1 homologues from the set of eukaryotic proteomes. Data S11 provides protein sequences of identified Mid1 homologues from the set of fungal proteomes. Data S12 provides protein sequences of identified UNC80 homologues from the set of eukaryotic proteomes. Data S13 provides protein sequences of identified UNC80 homologues from the set of fungal proteomes. Data S14 provides sequences of identified UNC79 homologues from the set of eukaryotic proteomes. Data S15 provides sequences of identified UNC79 homologues from the set of fungal proteomes. Data S16 provides protein sequences of identified NALCN homologues from animals. Data S17 provides protein sequences of identified FAM155 homologues from animals. Data S18 provides protein sequences of identified UNC80 homologues from animals. Data S19 provides protein sequences of identified UNC79 homologues from animals. Data S20 provides the raw phylogenetic tree, in nexus format, of Fig. 2 A. Data S21 provides the raw phylogenetic tree, in nexus format, of Fig. 2 B. Data S22 provides the raw phylogenetic tree, in nexus format, of Fig. 3 B. Data S23 provides the raw phylogenetic tree, in nexus format, of Fig. 3 C. Data S24 provides the raw phylogenetic tree, in nexus format, of Fig. 5 B. Data S25 provides the raw phylogenetic tree, in nexus format, of Fig. 5 C. Data S26 provides the raw phylogenetic tree, in nexus format, of Fig. 6 B. Data S27 provides the raw phylogenetic tree, in nexus format, of Fig. 6 C. Data S28 provides the raw phylogenetic tree, in nexus format, of Fig. S1. Data S29 provides the raw phylogenetic tree, in nexus format, of Fig. S2. Data S30 provides the raw phylogenetic tree, in nexus format, of Fig. S3. Data S31 provides the raw phylogenetic tree, in nexus format, of Fig. S4.

## Results

### Phylogeny of the NALCN and Cch1 subunits

Previous phylogenetic analyses demonstrated homology between NALCN and Cch1 channels (Liebeskind et al., 2012; Pozdnyakov et al., 2018). We sought to expand on this previous work by searching for homologues in a collection of 185 high-quality proteomes from a range of species spanning eukaryotes, utilizing a custom HMM strategy (i.e., with HMMer3), which is more sensitive than BLAST for identifying homologues (Madera and Gough, 2002). Thus, we generated an HMM profile trained on a set of manually selected NALCN and Cch1 channel protein sequences and used this to identify a total of 3,476 nonredundant protein sequences from our proteome collection. These were then filtered to remove ones with <18 predicted transmembrane helices, to exclude single and tandem domain pore-loop channels like voltage-gated potassium and two-pore channels, respectively. The

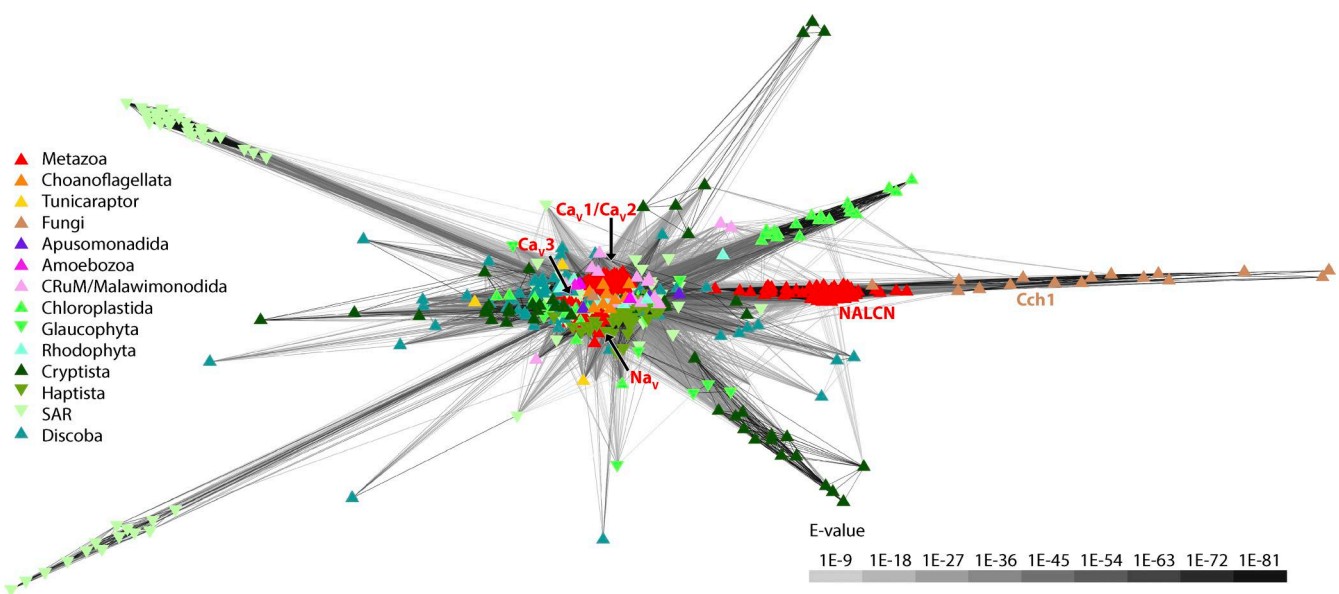

**Figure 1. Cluster map of identified eukaryotic pore-loop channels with ≥18 predicted transmembrane helices.** Edges correspond to BLAST comparison expect-values (E-values) colored according to the provided legend on the lower right of the map. Individual ion channel sequences are depicted by symbols that are colored according to the taxonomic groups indicated by the legend on the left of the plot. Metazoan NALCN, Na_V, and calcium (Ca_V) channels, as well as fungal Cch1 channels, are labeled.

resulting 1,476 candidate four-domain channel sequences were then analyzed with the all-against-all sequence similarity clustering program CLANS. This revealed a main cluster of 1,454 sequences, each with a minimum of three connections with other proteins, comprised of a large central group that includes metazoan Ca_V1/Ca_V2, Ca_V3, and Na_V channels, together with a diverse set of non-metazoan channels (Fig. 1). NALCN channels from animals, along with fungal Cch1 channels, formed a separate and highly connected subcluster, and several additional subclusters were evident for sets of channels from the stramenopile, alveolate, and rhizaria supergroup; Cryptista; and Chloroplastida. Of note, since our HMM trained on NALCN/Cch1 sequences was also able to detect other classes of distant four-domain channels, including Ca_V and Na_V, it seems unlikely that our search strategy missed detection of NALCN/Cch1 sequences within the utilized proteomes.

To explore the relationships of these HMM-identified channels, we generated a maximum likelihood phylogeny inferred from a trimmed protein sequence alignment. The resulting tree confirms the direct phylogenetic relationship between NALCN and Cch1 channels (Liebeskind et al., 2012; Pozdnyakov et al., 2018) and identifies a strongly supported sister clade relationship of NALCN/Cch1 with channels from species within Apusomonadida and Crumalia/CRuM (i.e., collodictyonids, rigifilids, and mantamonadids) (Fig. 2 A). Together, these associate with a larger set of channels from Discoba, Malawimonadida, and Cryptista, in a clade with variable node support, and a much broader strongly supported clade, which we refer to as clade A, with representation amongst eukaryotes. Metazoan Ca_V1 and Ca_V2 channels, Ca_V3 channels, and Na_V channels fall within three distant and separate clades, respectively (i.e., clades B, C, and D), distant from NALCN/Cch1 channels and indeed all clade

A channels (Fig. 2 A). All these three clades also contain other sets of channels from microbial eukaryotic taxa, including Choanoflagellata, Chloroplastida, Cryptista, and Apusomonadida.

Next, we used our NALCN/Cch1 HMM to search through an expanded set of 256 proteomes from the FungiDB genomic database, which includes species from all the major fungal lineages: Ascomycota, Basidiomycota, Blastocladiomycota, Chytridiomycota, and Mucoromycota (Basenko et al., 2018). This enabled us to identify Cch1 homologues in all lineages except Chytridiomycota (filtered to possess at least 18 predicted transmembrane helices). A phylogenetic tree of these protein sequences, rooted on human and *T. adhaerens* Ca_V and Na_V channels, reveals near congruency with the expected species phylogeny (Li et al., 2021), except for a switched branch position of Cch1 homologues from Pucciniomycotina and Ustilagomycotina within the larger taxonomic group of Basidiomycota (Fig. 2 B).

We also conducted a more detailed phylogenetic analysis of NALCN in animals, using a reciprocal BLAST search approach to identify homologues in gene data from representative species spanning most major bilaterian and non-bilaterian phyla (Fig. S1, inset). Except for ctenophores (comb jellies), we were able to identify NALCN homologues in all animal taxa. A phylogenetic tree of these protein sequences reveals most examined species possess only single gene copies of NALCN, with duplications apparent for the known *C. elegans* paralogues NCA-1 and NCA-2, the platyhelminth *Macrostomum lignano*, and the Great Barrier Reef sponge *A. queenslandica* (Fig. S1).

### FAM155 homologues are found in animals, fungi, apusomonads, and unicellular cryptist algae
Like Cch1, previous phylogenetic analyses established a phylogenetic link between FAM155 in animals and Mid1 in fungi

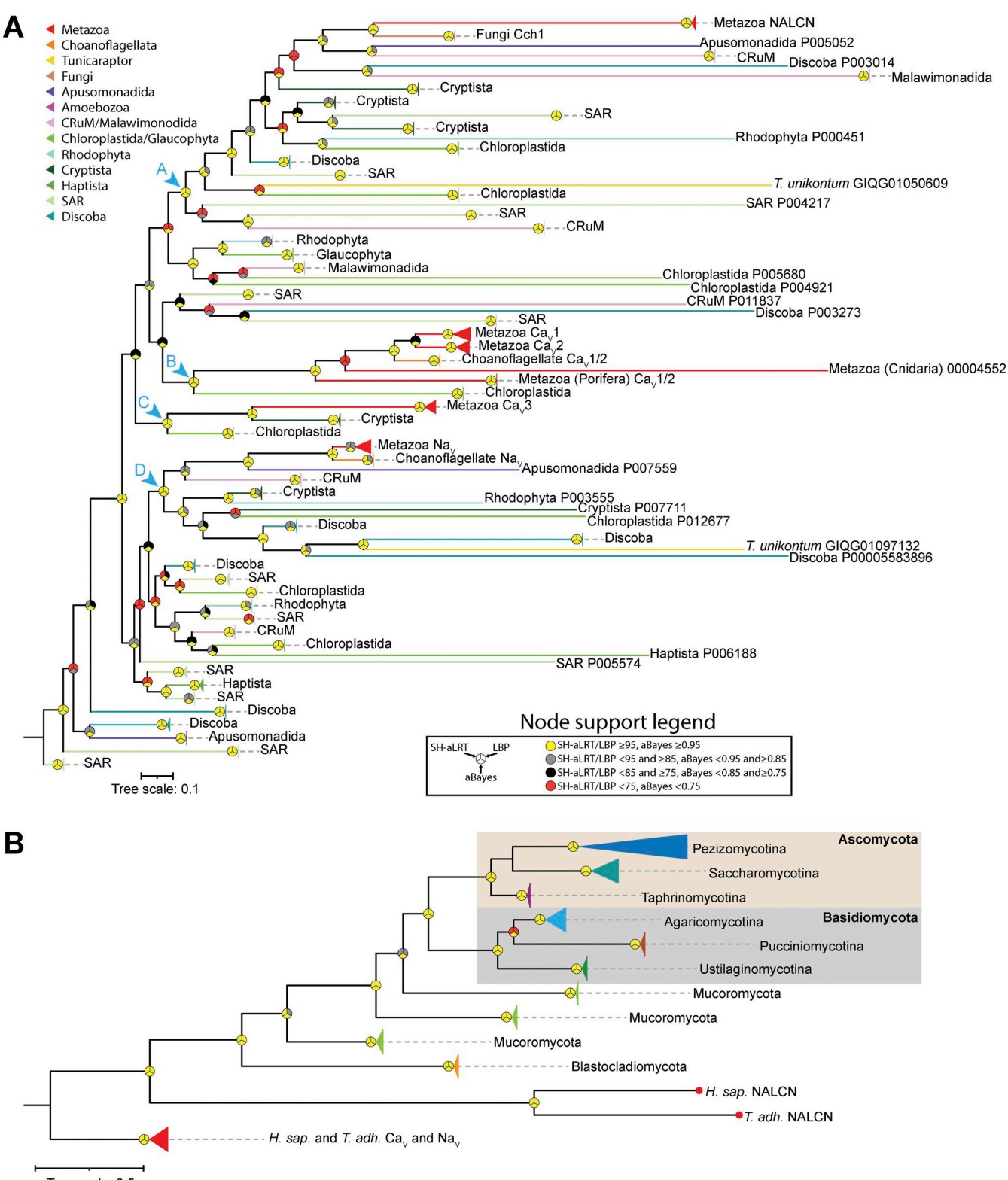

**Figure 2. Phylogenetic analysis of eukaryotic pore-loop channels related to NALCN and Cch1. (A)** Maximum likelihood phylogenetic tree of HMM-identified pore-loop channel protein sequences from a curated set of eukaryotic proteomes. Branches are colored according to taxonomic groupings as indicated by the legend. The chevrons with alphabetic labels (cyan colored) denote strongly supported nodes that separate NALCN/Cch1 (clade A), $Ca_V1/Ca_V2$ (clade B), $Ca_V3$ (clade C), and $Na_V$ (clade D) and associated eukaryotic channels from each other. **(B)** Maximum likelihood phylogenetic tree of select metazoan pore-loop channels and Cch1 homologues identified from an expanded set of fungal proteomes from the FungiDB database. Leaves on the tree are colored according to major taxonomic groupings within fungi. For both trees, node support values for three separate analyses, SH-aLRT, LBP, and aBayes, are depicted by circular symbols with colors reflecting ranges of values as indicated in the node support legend. SH-aLRT, Shimodaira–Hasegawa approximate likelihood ratio tests; aBayes, approximate Bayes tests; LBP, local bootstrap probability tests.

(Ghezzi et al., 2014). Using the same strategy that we used for NALCN and Cch1 channels, we generated an HMM trained on selected FAM155 and Mid1 sequences and used it to search through the set of 185 eukaryotic proteomes. This identified a total of 44 protein sequences, mostly from animals and fungi, as expected. However, as is evident in a sequence similarity cluster map, eight of these sequences were from algae-like species from the group Cryptista, with closer connections to fungal Mid1 sequences compared with FAM155 sequences from metazoans (Fig. 3 A). Of note, our HMM search failed to identify a previously identified FAM155/Mid1 homologue from the apusomonad species *Thecamonas trahens* (Ghezzi et al., 2014) (NCBI accession number XP_013754286.1), despite the sequence being present in the proteome we used in our analysis. We attribute this to the generally divergent nature of FAM155/Mid1 proteins (Ghezzi et al., 2014) and the atypical length of this protein of 2909 amino acids, compared, for example, to only 458 for human FAM155A and 623 for *C. neoformans* (fungal) Mid1 (see discussion). A maximum likelihood phylogeny based on the identified sequences reveals strong support for nodes separating metazoan, fungal, and cryptist FAM155/Mid1 homologues, but weak or absent support for most nodes within each taxon (Fig. 3 B). We used this same HMM model to search through our expanded set of fungal proteomes, identifying homologues in all lineages except Chytridiomycota, like the Cch1 subunit. Hence, it seems all fungal lineages that possess Cch1 also possess Mid1.

Like NALCN, searching for FAM155 sequences in animals via reciprocal BLAST allowed us to identify homologues in all major taxa except ctenophores. A phylogenetic tree based on these sequences indicates duplications occurred in vertebrates (i.e., FAM155A and FAM155B), the rotifer *Rotaria socialis*, and the platyhelminth *M. lignano* that also duplicated NALCN (Fig. S2).

Our identification of FAM155/Mid1 homologues in cryptists prompted us to explore their homology with metazoan FAM155 and fungal Mid1 proteins more deeply, along with the previously identified homologue from *T. trahens*. Using AlphaFold3 (Abramson et al., 2024), we predicted dimeric complexes of NALCN–FAM155 proteins from the early-diverging animal *T. adhaerens* and the Cch1-Mid1 homologues from the basidiomycete fungus *C. neoformans*, in which these have been shown to be functionally and genetically integrated (Liu et al., 2006; Hong et al., 2013; Vu et al., 2015). We also included in these analyses Mid1/FAM155 homologues from two cryptist species, *Cryptomonas curvata* and *Geminigera cryophila*, and the apusomonad *T. trahens*. Putative NALCN/Cch1 homologues for these species were selected from our phylogenetic analysis of four-domain channels, specifically, from the clades of cryptist and apusomonad channels most phylogenetically proximal to NALCN and Cch1 (Fig. 2 A) (Brown et al., 2018). Although such structural predictions must be interpreted with caution, we reasoned that AlphaFold could be used to explore whether complexing, in a manner consistent with the known structures of human NALCN and FAM155, is at least possible for these various proteins.

Unfortunately, AlphaFold failed to predict the dimeric structure of the *T. trahens* NALCN/Cch1 and FAM155/Mid1 proteins, producing a low pTM score of 0.45 (all scores and accession numbers for our structural predictions are provided in

Table S2). Nonetheless, all other examined dimers produced acceptable scores above 0.5, all containing a set of α1 to α3 helices positioned atop the channel (Fig. 4, A and B), which in the solved human complex mediates critical contacts with the NALCN subunit (Kang et al., 2020; Kschonsak et al., 2020; Xie et al., 2020). We also predicted the structure of the human NALCN–FAM155A complex and structurally aligned it with its corresponding solved structure (Kschonsak et al., 2022), revealing highly overlapping structures with a root mean square alignment deviation score of 0.784 Å (Fig. S5). Outside of the α1 to α3 helices, there is marked variation in the predicted structures of FAM155/Mid1 proteins, with a set of β sheets in the *T. adhaerens* subunit, an extended globular arrangement in the *C. neoformans* subunit, and various flanking alpha helices in the *G. cryophila* subunit. Notably, the *T. adhaerens* FAM155 subunit is also predicted to possess a hydrophobic alpha helix that runs alongside the NALCN subunit (Fig. 4 A). Versions of these predicted structures, colored according to pLDDT confidence scores, are provided in Fig. S6 A.

A protein alignment of human FAM155A with homologues from *T. adhaerens*, the fungi *C. neoformans* and *Rhizophagus irregularis*, the apusomonad *T. trahens*, and the cryptists *C. curvata* and *G. cryophila* reveals conserved sites in α1 to α3 helices, within otherwise highly divergent protein sequences (Fig. 4 C). Among the three helices, α1 is the most divergent, both in terms of sequence and length (i.e., 16 to 38 amino acids). The α2 and α3 helices exhibit more conservation, most possessing highly conserved cysteine residues that serve to stabilize the tertiary arrangement of the α1 to α3 helices in resolved structures (Kang et al., 2020; Kschonsak et al., 2020; Xie et al., 2020). Several amino acids within and around these alpha helices form critical contacts with NALCN in mammals, but our alignment shows that most of these are not conserved. The exceptions are a tyrosine-serine motif in the α1–α2 linker, a tyrosine residue in α2, and a cysteine in α3. Thus, it appears considerable changes have occurred in molecular determinants that mediate the interaction between NALCN/Cch1 and FAM155/Mid1 subunits. Also notable is that the region located downstream of α3 (i.e., the post α3 loop), which is believed to be essential for interactions with NALCN in mammals, is poorly conserved in non-vertebrate sequences beyond a ubiquitously conserved cysteine-proline motif at the very start of this linker (Fig. 4 C).

## UNC79 and UNC80 homologues are found within several distant eukaryotic lineages

To date, unlike NALCN/Cch1 and FAM155/Mid1, bona fide UNC79 and UNC80 homologues have not been identified in either fungi or any other non-metazoan eukaryotes. Nonetheless, given the expanded set of gene sequences available for fungi and other eukaryotes, we sought to reexamine the presence of these two genes outside of animals. A BLAST search through the NCBI nonredundant database identified putative UNC80 homologues from several fungal species, including *Rhizopus delemar* (Mucoromycota; NCBI accession number KAG1050648.1). A reciprocal BLAST of this sequence against metazoan NCBI sequences identified UNC80 homologues as top hits, and global alignment of this protein with the human UNC80 sequence revealed a low percent identity of 22.3% and percent similarity of 37.1% (Data S1).

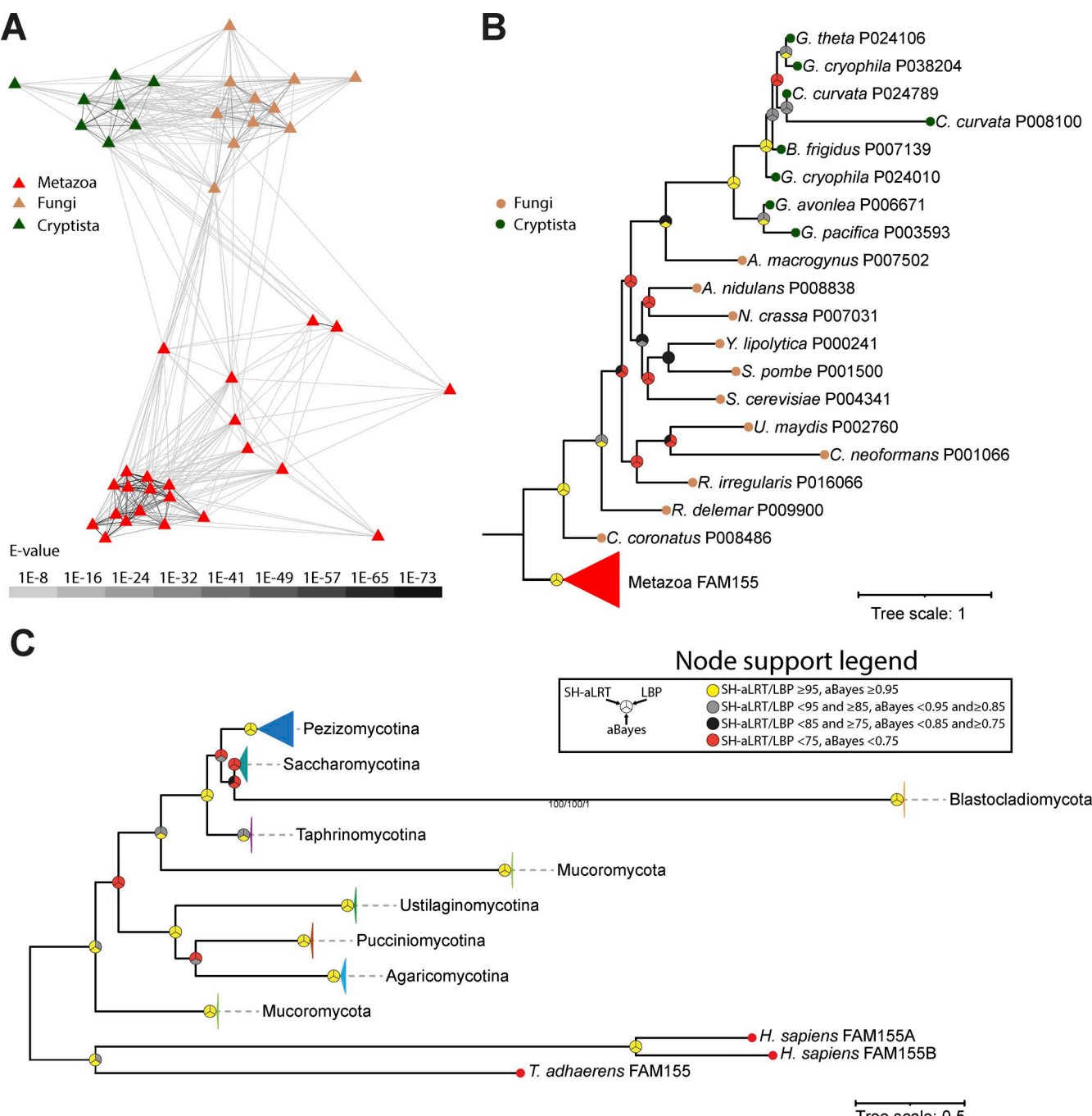

Figure 3. **FAM155/Mid1 homologues are found in Metazoa, Fungi, and Cryptista. (A)** Cluster map of identified FAM155/Mid1 protein sequences from eukaryotes. Edges correspond to BLAST comparison E-values colored according to the legend. **(B)** Maximum likelihood tree of HMM-identified FAM155/Mid1 homologues in eukaryotes. **(C)** Maximum likelihood tree of HMM-identified FAM155/Mid1 homologues in an expanded set of fungal proteomes. Leaves on the tree are colored according to major taxonomic groupings within fungi. For both trees, node support values for three separate analyses, SH-aLRT, LBP, and aBayes, are depicted by circular symbols with colors reflecting ranges of values as indicated in the node support legend. SH-aLRT, Shimodaira–Hasegawa approximate likelihood ratio tests; aBayes, approximate Bayes tests; LBP, local bootstrap probability tests; E-values, expect-values.

With a stringent expect-value cutoff of 1E-30, we used the *R. delemar* sequence to extract additional homologues from the FungiDB database and combined these with a set of metazoan UNC80 homologues to train a HMM for searching through our set of eukaryotic proteomes. This led to the identification of 139 UNC80 homologues, most from metazoans, but also from fungi, Rotosphaerida, Apusomonadida, Malawimonadida, CRuMs, and

Discoba (Fig. 5 A). A phylogeny of these proteins reveals strong support for the separation of animal UNC80 sequences from those of other eukaryotes and respective monophyletic relationships among the five fungal sequences and the two from Discoba.

To better examine the presence of UNC80 in fungi, we generated two separate HMMs, one of combined metazoan and fungal sequences and another with just fungal sequences, and

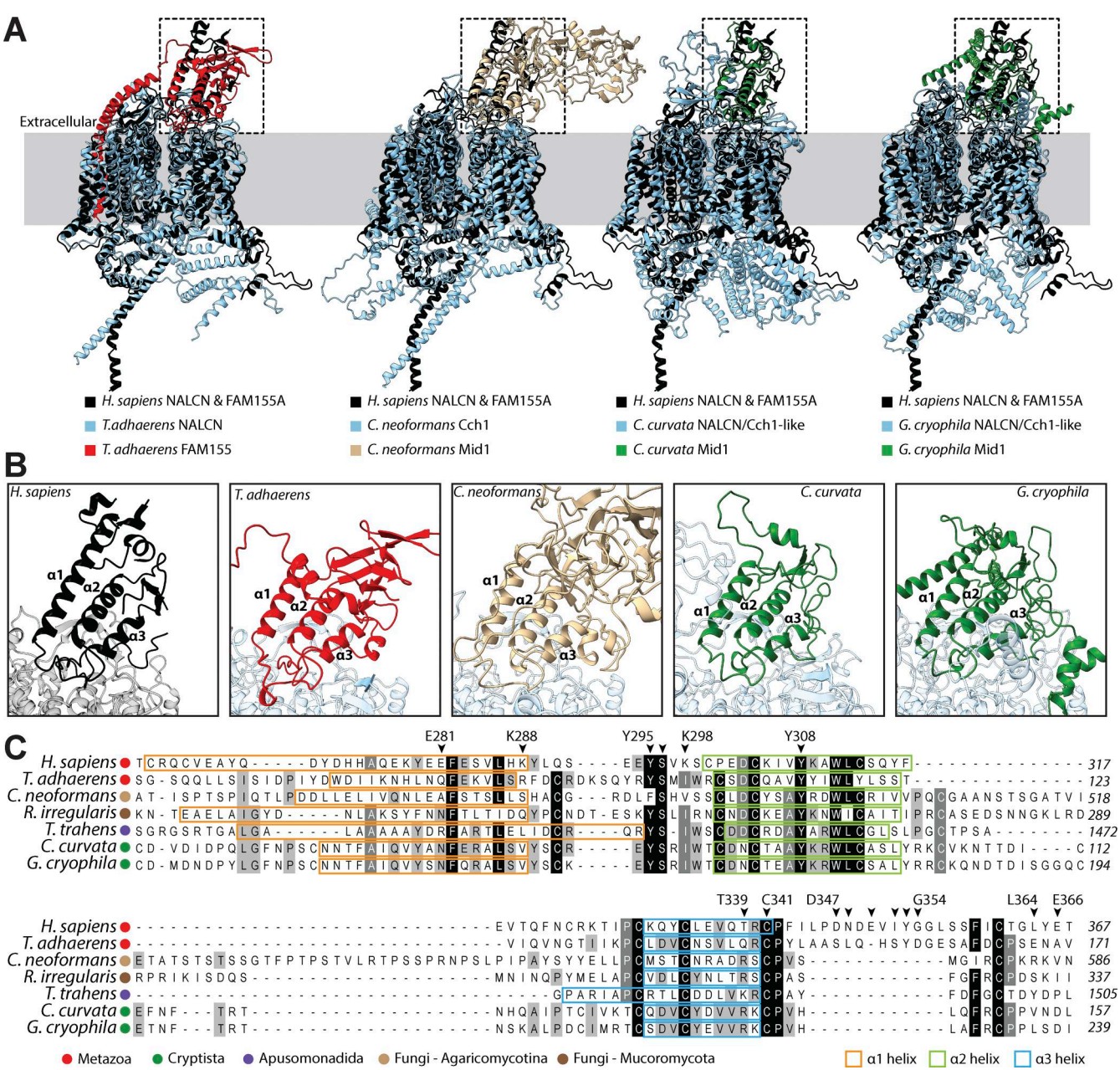

Figure 4. **Structural and sequence analysis of FAM155/Mid1 homologues. (A)** Structural superimposition of the solved human NALCN–FAM155A cryo-EM structure (from PDB accession no. 7SX4) and predicted NALCN–FAM155 or Cch1-Mid1 dimer structures from the early-diverging animal *T. adhaerens*, the pathogenic fungus *C. neoformans*, and the Cryptista (algal) species *C. curvata* and *G. cryophila*. **(B)** Close-up view of the FAM155/Mid1 NALCN/Cch1 subunit interfaces (corresponding to the dashed black boxes in panel A, revealing similar positioning of predicted α1 to α3 helices, that in the human complex, form critical contacts with the NALCN subunit. Versions of these predicted structures, colored according to pLDDT confidence scores, are provided in Fig. S6 A. **(C)** Partial protein alignment of selected FAM155/Mid1 homologues from animals, fungi, apusomonads, and cryptists, revealing strong sequence conservation within the α1 to α3 helices. The chevrons above the alignment denote amino acids in the human FAM155A shown to make important contacts with NALCN in cryo-EM structures. Numbers to the right of the alignment denote amino acid positions. The boxes are used to label amino acids predicted to form helical structures by AlphaFold.

used these to search through the set of FungiDB proteomes. While both models identified a common set of 42 candidate homologues, the fungi-only model identified two additional sequences. Thus, we selected this latter set of 44 sequences for phylogenetic analysis. A maximum likelihood tree, rooted on the human and *T. adhaerens* UNC80 homologues, reveals narrower representation among fungal species compared with Cch1 and

Mid1, restricted to agaricomycetes and tremellomycetes within the Basidiomycota and Agaricomycotina, and glomeromycetes and mucoromycetes within Mucoromycota. Focusing on animals using the reciprocal BLAST strategy, we were able to identify UNC80 homologues in all examined species except ctenophores, thus similar to NALCN and FAM155, with an apparent duplication of this subunit in *R. socialis* (Fig. S6).

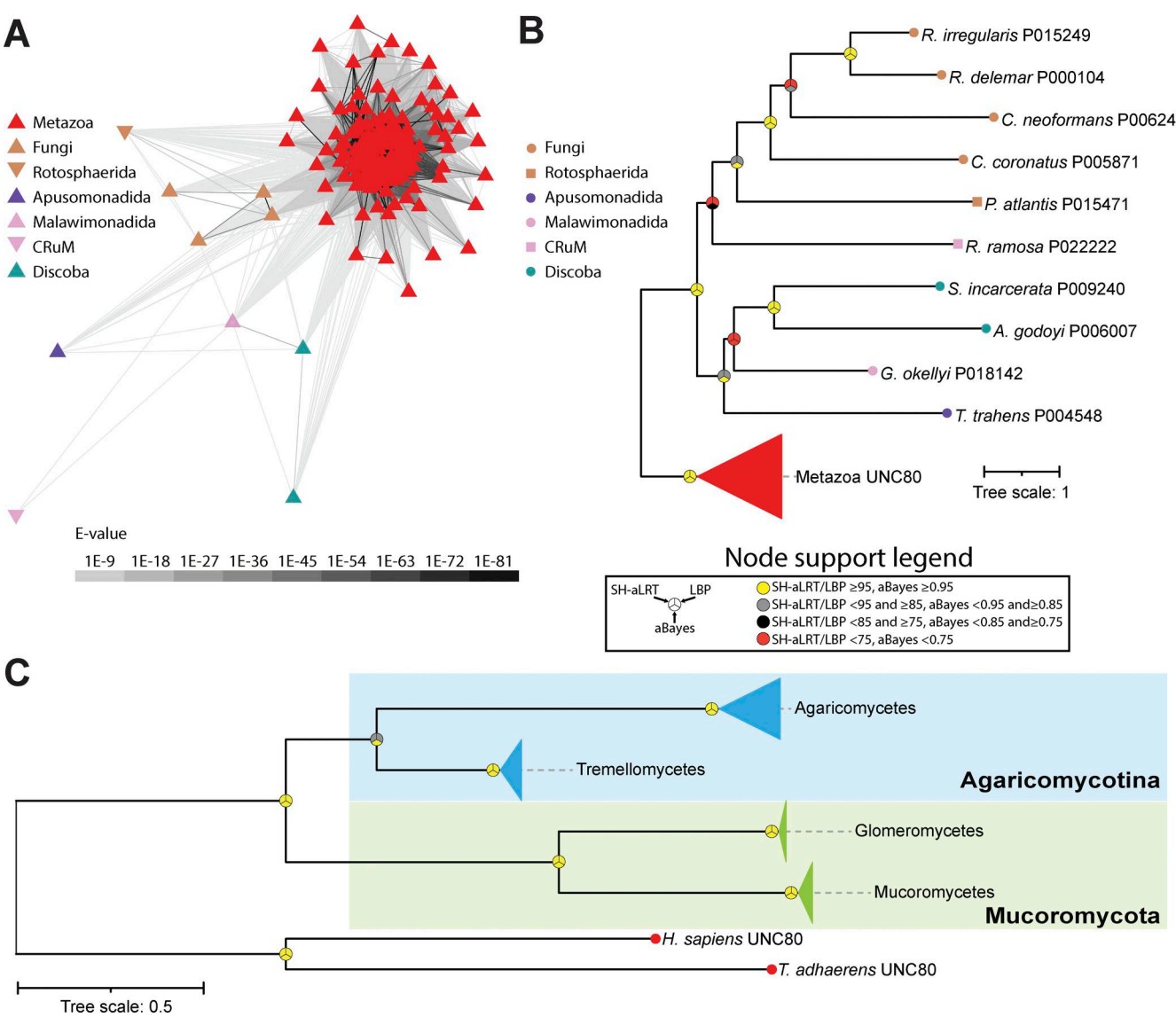

**Figure 5. Identification of UNC80 homologues in fungi and other eukaryotes. (A)** Cluster map of identified UNC80 protein sequences from eukaryotes. Edges correspond to BLAST comparison E-values colored according to the legend. **(B)** Maximum likelihood tree of HMM-identified UNC80 homologues in eukaryotes. **(C)** Maximum likelihood tree of HMM-identified UNC80 homologues in an expanded set of fungal proteomes. Leaves on the tree are colored according to major taxonomic groupings within fungi. For both trees, node support values for three separate analyses, SH-aLRT, LBP, and aBayes, are depicted by circular symbols with colors reflecting ranges of values as indicated in the node support legend. SH-aLRT, Shimodaira–Hasegawa approximate likelihood ratio tests; aBayes, approximate Bayes tests; LBP, local bootstrap probability tests; E-values, expect-values.

Unlike UNC80, a BLAST search for UNC79 homologues outside of animals failed to produce any hits in the NCBI nonredundant database. However, while conducting these analyses, a preprint article was released documenting the identification of putative UNC79 and UNC80 homologues in the basidiomycete *C. neoformans*. Through large-scale genotype–phenotype clustering, this study found a phenotypic link between the previously characterized Cch1 and Mid1 genes and two previously uncharacterized and large proteins bearing predicted armadillo repeats, a feature of UNC79 and UNC80 (Boucher et al., 2024, *Preprint*). This study also demonstrated a biochemical interaction between these two proteins and both Cch1 and Mid1, altogether suggesting these form a complex with Cch1 and are homologous to the animal UNC79 and UNC80 subunits of NALCN

(Boucher et al., 2024, *Preprint*). However, phylogenetic evidence supporting this homology was not generated.

Using the putative *C. neoformans* UNC79 homologue, we identified several additional fungal sequences via a BLAST search of the FungiDB database (using an expect-value cutoff of 1E-30) and combined a selection of these with metazoan UNC79 sequences for training an HMM and used this combined model to search for homologues in our eukaryotic databases. This identified a total of 140 sequences, which, interestingly, were similar to UNC80 in terms of species composition, most coming from animals and fungi, but also species from Apusomonadida, Malawimonadida, and Discoba (Fig. 6, A and B). Our inability to detect fungal UNC79 sequences via BLAST, using animal UNC79 sequences as queries, indicates strong sequence divergence. A

global alignment of the *C. neoformans* and human UNC79 proteins yields only 23.1% sequence identity and 37.8% sequence similarity, distributed broadly along the length of the alignment without any extended stretches of matching sequence (Data S1). This is different from our alignment of the *R. delemar* and human UNC80 (Data S1) and likely accounts for the inability of BLAST to seed and extend a high-scoring pair based on metazoan UNC79 query sequences. Considering this, we decided to generate an HMM of UNC79 based exclusively on the manually selected fungal sequences from our combined model and used this to search for homologues in the FungiDB data sets. This yielded a total of 39 hits, which, when analyzed phylogenetically, produced a tree very similar to the fungal UNC80 phylogeny with respect to species composition and topology (Fig. 5 C and Fig. 6 C). Evident from this analysis is that, like UNC80, UNC79 homologues are restricted to agaricomycetes and tremellomycetes within the Agaricomycotina and glomeromycetes and mucoromycetes within the Mucoromycota. Together, these analyses indicate that select clades of fungi within Agaricomycotina and Mucoromycota possess a complete set of NALCN channelosome subunits: NALCN/Cch1, FAM155/Mid1, UNC79, and UNC80. Lastly, reciprocal BLAST and phylogenetic analysis of UNC79 homologues, focused strictly within animals, identified sequences in all examined species except ctenophores (Fig. S4), altogether indicating that all animals, except ctenophores, possess a complete set of NALCN subunits.

Clearly, both the identified non-metazoan UNC79 and UNC80 homologues are highly divergent in their protein sequence relative to their metazoan counterparts. In cryo-EM structures, the human subunits both form longitudinal, S-shaped proteins, each composed of >30 armadillo repeats, which together form a subdimer in an inverted N- to C-terminal supercoiled orientation (Kang and Chen, 2022; Kschonsak et al., 2022; Zhou et al., 2022) (Fig. 7 A). We sought to predict whether such configurations are possible for the identified non-metazoan homologues using AlphaFold3. Specifically, we predicted the dimeric structures of homologues from the species *T. adhaerens*, *C. neoformans* (fungi, Agaricomycotina), *R. irregularis* (fungi, Mucoromycota), *T. trahens* (Apusomonadida), *Gefionella okellyi* (Malawimonadida), and *Stygiella incarcerata* (Discoba). All proteins were predicted to form armadillo repeat proteins, with dimers forming inverted N- to C-terminal arrangements. However, only the homologues from *T. adhaerens*, *C. neoformans*, *R. irregularis*, *T. trahens*, and *G. okellyi* had linear supercoiled structures that could be structurally aligned with the solved structures of human UNC79 and UNC80 (Fig. 7, B–F). Instead, the predicted dimer from *S. incarcerata* lacked supercoiling (Fig. 7 G). Versions of these predicted structures, colored according to pLDDT confidence scores, are provided in Fig. S6 B. Together, the structural predictions are consistent with the ability of these proteins to form dimeric complexes like the known structure of human UNC79 and UNC80, further support the homology of these non-metazoan proteins to the metazoan NALCN ancillary subunits. Furthermore, they are consistent with the recent biochemical evidence that the UNC79 and UNC80 homologues from *C. neoformans* form a functional and physical complex with Cch1 and Mid1 (Boucher et al., 2024, *Preprint*).

## Discussion

Our phylogenetic analysis of four-domain channels identified using a custom HMM profile built from aligned NALCN and Cch1 protein sequences revealed that NALCN, $Ca_V1$/$Ca_V2$, $Ca_V3$, and $Na_V$ channels fall into four separate and strongly supported clades (i.e., clades A to E), each comprised of additional nonmetazoan channels from a broad range of eukaryotes (Fig. 2 A). In clade A, which included NALCN and Cch1, there were also channels from Apusomonadida, Malawimonadida, and Crumalia/CRuMs, which formed phylogenetic relationships with NALCN/Cch1 roughly consistent with the species phylogeny (Brown et al., 2018) (Fig. 8). Clade A also contained more divergent channels from eukaryotes within the Diaphoretikes (i.e., Chloroplastida; Glaucophyta; Rhodophyta; Cryptista; stramenopile, alveolate, and rhizaria supergroup; and Discoba), indicating that NALCN and Cch1 belong to a large and ancient clade of four-domain channels, conserved between Amorphea and Diaphoretikes, that is distinct from $Ca_V$ and $Na_V$ channels.

Previous studies demonstrated orthologous phylogenetic relationships between metazoan NALCN and Mid1 proteins and fungal Cch1 and Mid1 proteins, respectively (Liebeskind et al., 2012; Ghezzi et al., 2014). Here, we extend these observations by demonstrating that two separate lineages of fungi, Agaricomycotina within the larger clade Basidiomycota and Mucoromycota, also possess UNC79 and UNC80 homologues (Fig. 8). We also found homologues of UNC79 and UNC80 more broadly in eukaryotes, within several non-opisthokont lineages within Amorphea and Discoba within Diaphoretikes (Fig. 8). Structural predictions of sets of these proteins from various representative species corroborated their homology to metazoan UNC79 and UNC80, most arranged as inverted N- to C-terminal supercoiled quaternary structures with each subunit made of repeating armadillo repeats (Fig. 7), similar to the solved structures of human UNC79 and UNC80 in complex with NALCN and FAM155 (Kang and Chen, 2022; Kschonsak et al., 2022; Zhou et al., 2022) (Fig. 8). That these are true homologues of metazoan UNC79 and UNC80 is supported by recent biochemical experiments revealing that Cch1, Mid1, UNC79, and UNC80 from the fungal species *C. neoformans* (Agaricomycota) form a physical complex *in vitro* and large-scale mutation analysis indicating the four proteins form a functional phenotypic module *in vivo* (Boucher et al., 2024, *Preprint*).

Outside of animals and fungi, the least prevalent subunit we found in our analysis was FAM155/Mid1 (Fig. 8). However, we did identify bona fide homologues from Cryptista, with conserved α1 to α3 helices that form a core structural scaffold important for interactions with the NALCN subunit in solved structures of the mammalian channel complex (Kang et al., 2020; Kschonsak et al., 2020; Xie et al., 2020) (Fig. 4). Worth noting is that FAM155/Mid1 proteins are relatively short in sequence and highly divergent outside of the α1 to α3 helices, and as such, our HMM search might have failed to identify some homologues. Indeed, this was the case for the FAM155/Mid1 homologue from the apusomonad *T. trahens* that was identified in a previous study (Ghezzi et al., 2014). Perhaps, our HMM failed to identify this homologue not only because of sequence divergence, but also because of its unique extended length of

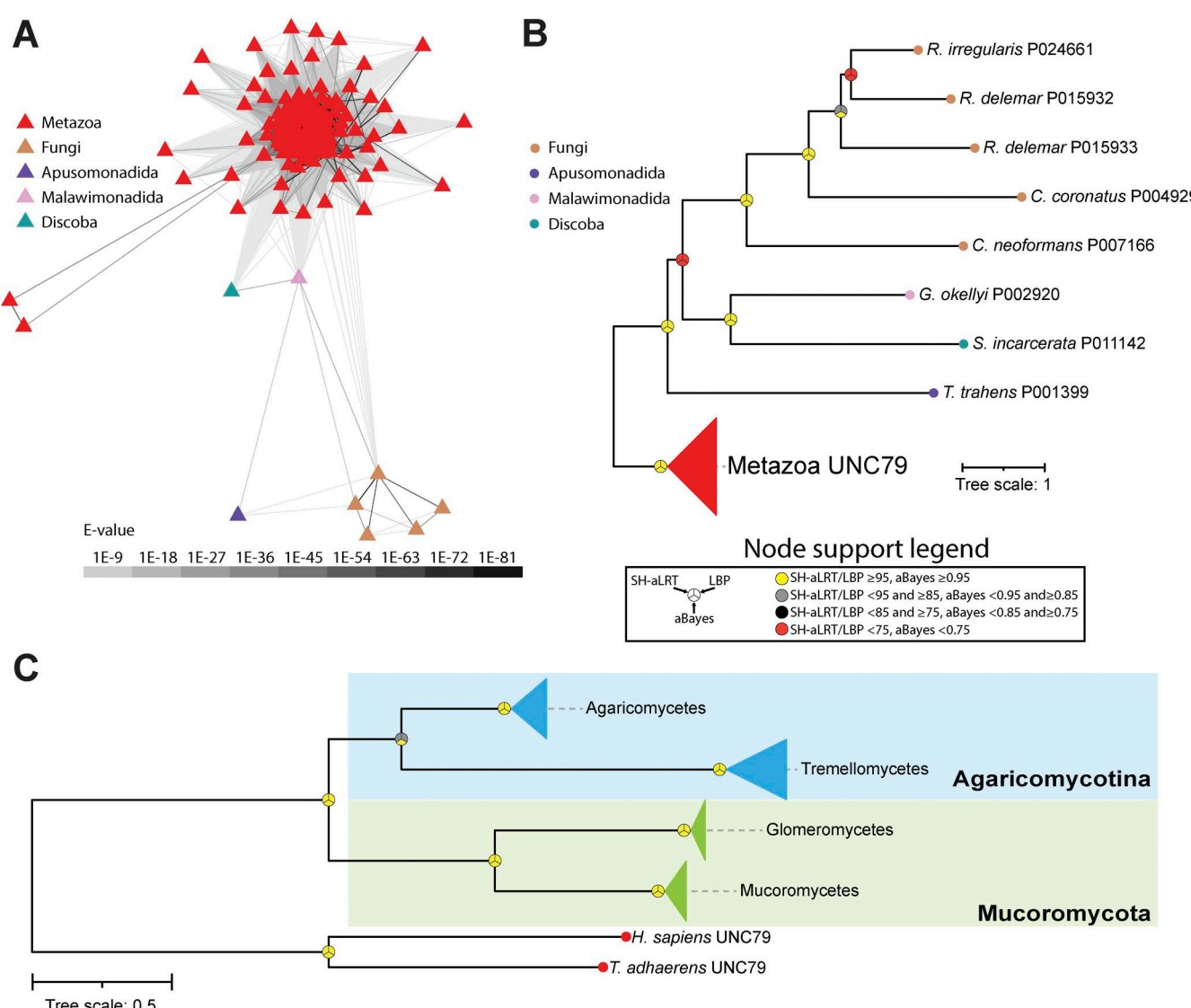

**Figure 6. Identification of UNC79 homologues in fungi and other eukaryotes. (A)** Cluster map of identified UNC79 protein sequences from eukaryotes. Edges correspond to BLAST comparison E-values colored according to the legend. **(B)** Maximum likelihood tree of HMM-identified UNC79 homologues in eukaryotes. **(C)** Maximum likelihood tree of HMM-identified UNC79 homologues in an expanded set of fungal proteomes. Leaves on the tree are colored according to major taxonomic groupings within fungi. For both trees, node support values for three separate analyses, SH-aLRT, LBP, and aBayes, are depicted by circular symbols with colors reflecting ranges of values as indicated in the node support legend. SH-aLRT, Shimodaira–Hasegawa approximate likelihood ratio tests; aBayes, approximate Bayes tests; LBP, local bootstrap probability tests; E-values, expect-values.

2,909 amino acids, compared with the much shorter sequences for FAM155/Mid1 proteins we identified for animals, fungi, and cryptists (Fig. 4 C; and Data S10 and S11). Likewise, armadillo repeat proteins like UNC79 and UNC80 tend to be highly divergent, beyond retaining conserved helical tertiary structures (Tewari et al., 2010). Thus, it may be that sequences were missed, such that future analyses with improved taxon sampling combined with novel approaches might uncover additional FAM155/Mid1, UNC79, and UNC80 homologues in other eukaryotic lineages.

However, despite possible omissions, our combined analyses point to an ancient origin of the complete NALCN/Cch1 channelosome complex, with conservation of all four subunits in at least three eukaryotic lineages: metazoans, fungi, and

apusomonads, all within Amorphea (Fig. 8). In animals, for which our search was more extensive and directed, we found homologues of all NALCN channelosome subunits in all examined taxa except ctenophores (comb jellies), proposed to be the earliest diverging metazoans (Dunn et al., 2008; Schultz et al., 2023). Similarly in fungi, our deep search found Cch1 homologues in most phyla, all of which were also found to possess Mid1. However, as noted, only species from Mucoromycota and Agaricomycotina within Basidiomycota were found to possess UNC79 and UNC80 (Fig. 8). Examining the presence/absence of genes strictly within Opisthokonta, and assuming the absences are real, it seems all NALCN/Cch1 subunits were lost in the intervening lineages between animals and fungi, outside of a single channel found for the holozoan *Tunicaraptor unikontum*.

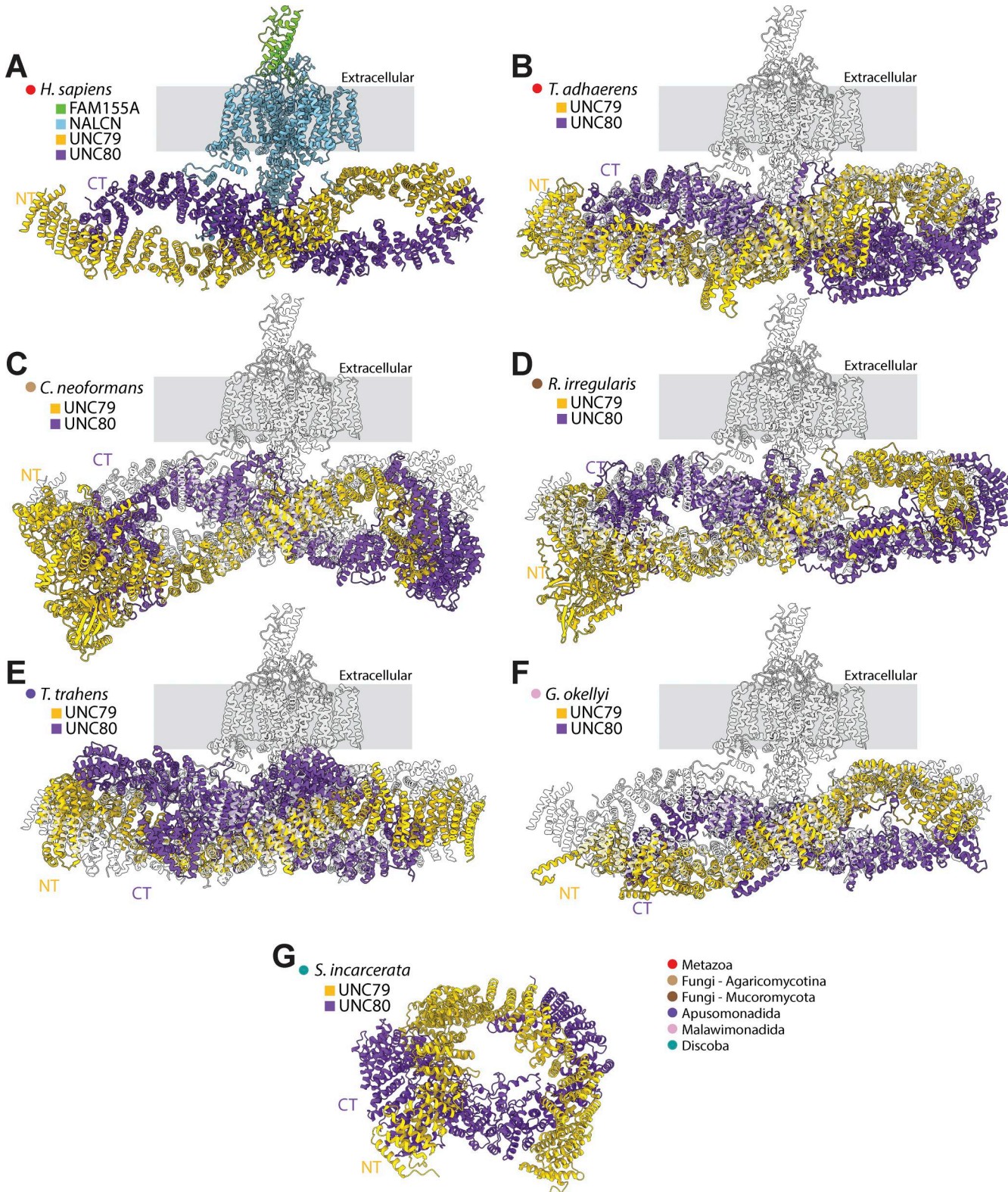

Figure 7. **Structural analysis of UNC79 and UNC80 homologues from animals and other eukaryotes. (A)** Cryo-EM structure of the human NALCN channel complexed with the subunit FAM155A and the cytoplasmic subunits UNC79 and UC80 (PDB accession number 7SX4). **(B)** Structural superimposition of the human NALCN complex (in semitransparent white) with the predicted structure of the UNC79/UNC80 dimer from the early-diverging metazoan *T. ad-haerens*. **(C)** Structural superimposition of the human NALCN complex with the predicted structure of the UNC79/UNC80 dimer from the fungal species *C. neoformans*. **(D)** Structural superimposition of human NALCN complex with the predicted structure of the UNC79/UNC80 dimer from the fungal species *R. irregularis*. **(E)** Structural superimposition of the human NALCN complex with the predicted structure of the UNC79/UNC80 dimer from the apusomonad *T. trahens*. **(F)** Structural superimposition of the human NALCN complex with the predicted structure of the UNC79/UNC80 dimer from the malawimonad

*G. okellyi.* **(G)** Predicted structure of the UNC79/UNC80 dimer from the Discoba species *S. incarcerata*. The legend in the lower right indicates relevant taxonomic groupings. For all panels, the labels NT and CT denote the rough location of the N and C termini of UNC79 and UNC80 proteins, respectively, along a horizontal plane. Versions of these predicted structures, colored according to pLDDT confidence scores, are provided in Fig. S6 B.

Within fungi, UNC79 and UNC80 appear to have been independently lost in Ascomycota, Ustilagomycota, Pucciniomycota, and Blastocladiomycota, while Chytridiomycota lost the entire Cch1 channelosome complex (Fig. 8).

Interestingly, the complete co-occurrence of all four NALCN subunits within animals suggests they have an obligate functional relationship with each other, which is consistent with the various genotype–phenotype studies that have been done in nematode worms, fruit flies, and mice (reviewed in Ren [2011], Cochet-Bissuel et al. [2014], Monteil et al. [2024]). In contrast, many lineages of fungi possess Cch1 and Mid1 but lack UNC79 and UNC80 (Fig. 8). Extensive work in the brewer's yeast *Saccharomyces cerevisiae* (Saccharomycotina) has revealed that Cch1 and Mid1 interact and function together (Viladevall et al., 2004; Peiter et al., 2005; Hayashi et al., 2020), and the absence of

UNC79 and UNC80 in this species indicates these two subunits can operate without the large cytoplasmic subunits, at least in some fungi. Furthermore, Cch1 and Mid1 from the basidiomycete fungus *C. neoformans* were reported to form functional $Ca^{2+}$ channels *in vitro* (Hong et al., 2010; Hong et al., 2013), although this species possesses UNC79 and UNC80 (Fig. 7), contrasting recent work on mammalian channels expressed *in vitro*, where reconstitution of functional currents required co-expression of all four subunits (Bouasse et al., 2019; Chua et al., 2020). These observations raise interesting questions about the conserved or divergent functions of UNC79, UNC80, and indeed all NALCN/Cch1 subunits, within different organismal lineages. Additional questions surround the conservation of molecular contacts between subunits that are required for complex assembly. Finding answers to these questions will require functional and structural

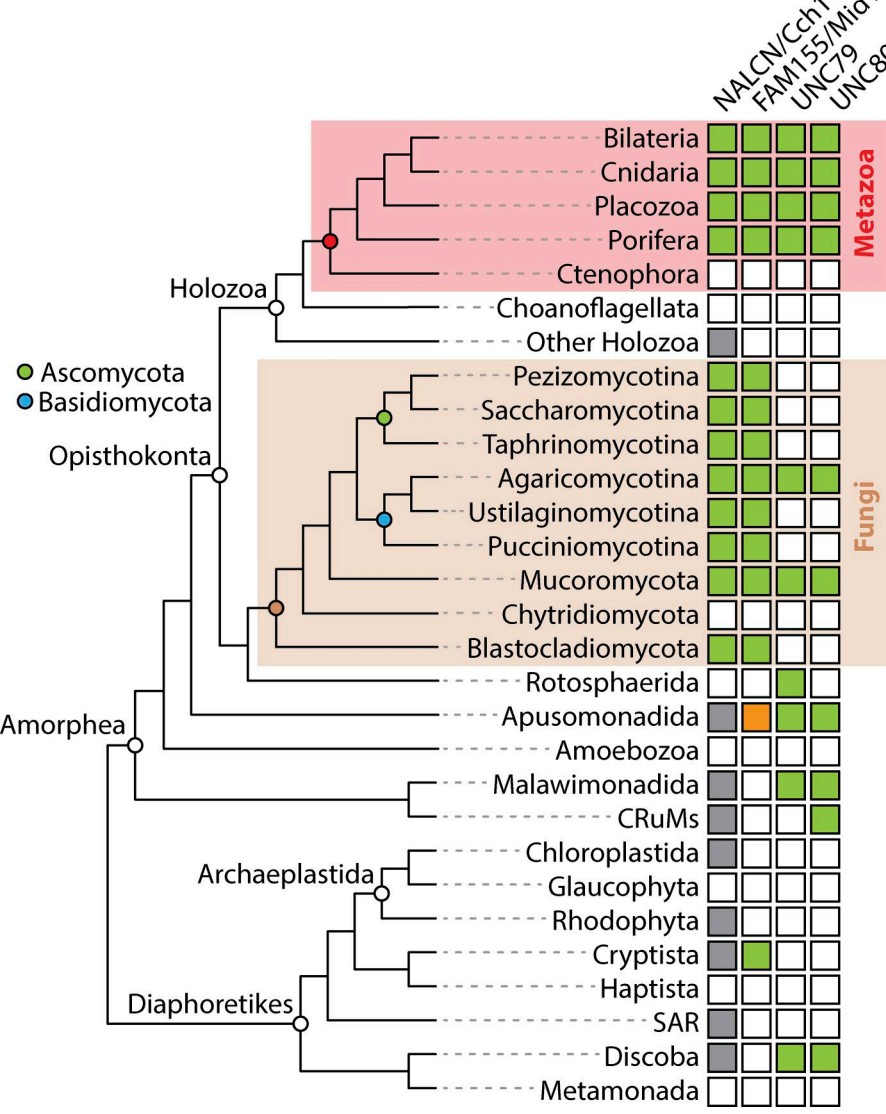

Figure 8. **Summary of identified NALCN/Cch1 channelosome subunits in eukaryotes.** The presence of homologues identified in this study is indicated by green colored boxes, with taxonomic groupings according to recent phylogenomic studies (Li et al., 2021; Strassert et al., 2021; Al Jewari and Baldauf, 2023; Schultz et al., 2023). Boxes colored in gray indicate the presence of NALCN/Cch1-related channels within clade A of our phylogenetic analysis shown in Fig. 2 A. The box colored in orange represents a FAM155/Mid1 homologue previously identified for the apusomonad *T. trahens* (Ghezzi et al., 2014).

characterization of these channel complexes from an expanded set of eukaryotic organisms, including invertebrate animals (Senatore et al., 2013; Boone et al., 2014). Such comparative work, and the establishment of a diversity of organisms for studying NALCN/Cch1 function, has the potential to uncover important insights into the molecular physiology of these channels and their subunits within a broad eukaryotic context.

## Data availability

Data pertaining to all figures are provided in the published article and its online supplemental material.

## Acknowledgments

Crina M. Nimigean served as editor.

This research was funded by an NSERC Discovery Grant (RGPIN-2021-03557), an NSERC Discovery Accelerator Supplement (RGPAS-2021-00002), and funds from the University of Toronto Mississauga Office of the Vice-Principal, Research and Innovation to Adriano Senatore; an Agence Nationale de la Recherche (ANR-21-NEU2-0004-01) to Arnaud Monteil; and an Ontario Graduate Scholarships to Wassim Elkhatib.

Author contributions: Adriano Senatore: conceptualization, data curation, formal analysis, funding acquisition, investigation, project administration, resources, supervision, validation, visualization, and writing—original draft, review, and editing. Tatiana D. Mayorova: investigation, visualization, and writing—review and editing. Luis A. Yañez-Guerra: data curation, investigation, and writing—review and editing. Wassim Elkhatib: data curation, software, and writing—review and editing. Brian Bejoy: data curation, software, and visualization. Philippe Lory: conceptualization, funding acquisition, resources, supervision, validation, and writing—review and editing. Arnaud Monteil: conceptualization, funding acquisition, project administration, supervision, visualization, and writing—original draft, review, and editing.

Disclosures: The authors declare no competing interests exist.

Submitted: 8 July 2024

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

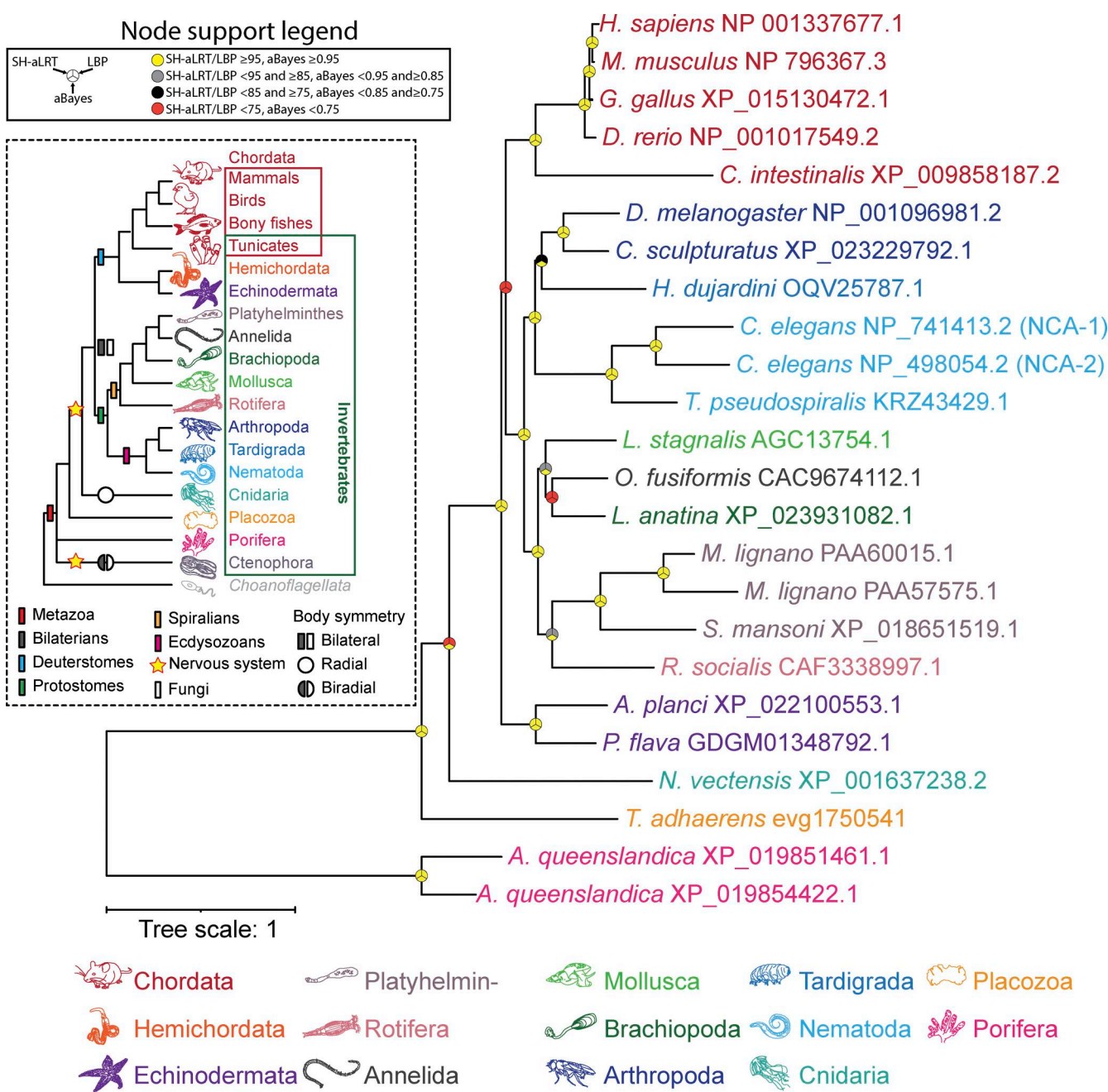

Figure S1. **Maximum likelihood tree of manually identified NALCN homologues from animals.** Node support values for three separate analyses, SH-aLRT, LBP, and aBayes, are depicted by circular symbols with colors reflecting ranges of values as indicated in the legend. SH-aLRT, Shimodaira–Hasegawa approximate likelihood ratio tests; aBayes, approximate Bayes tests; LBP, local bootstrap probability tests.

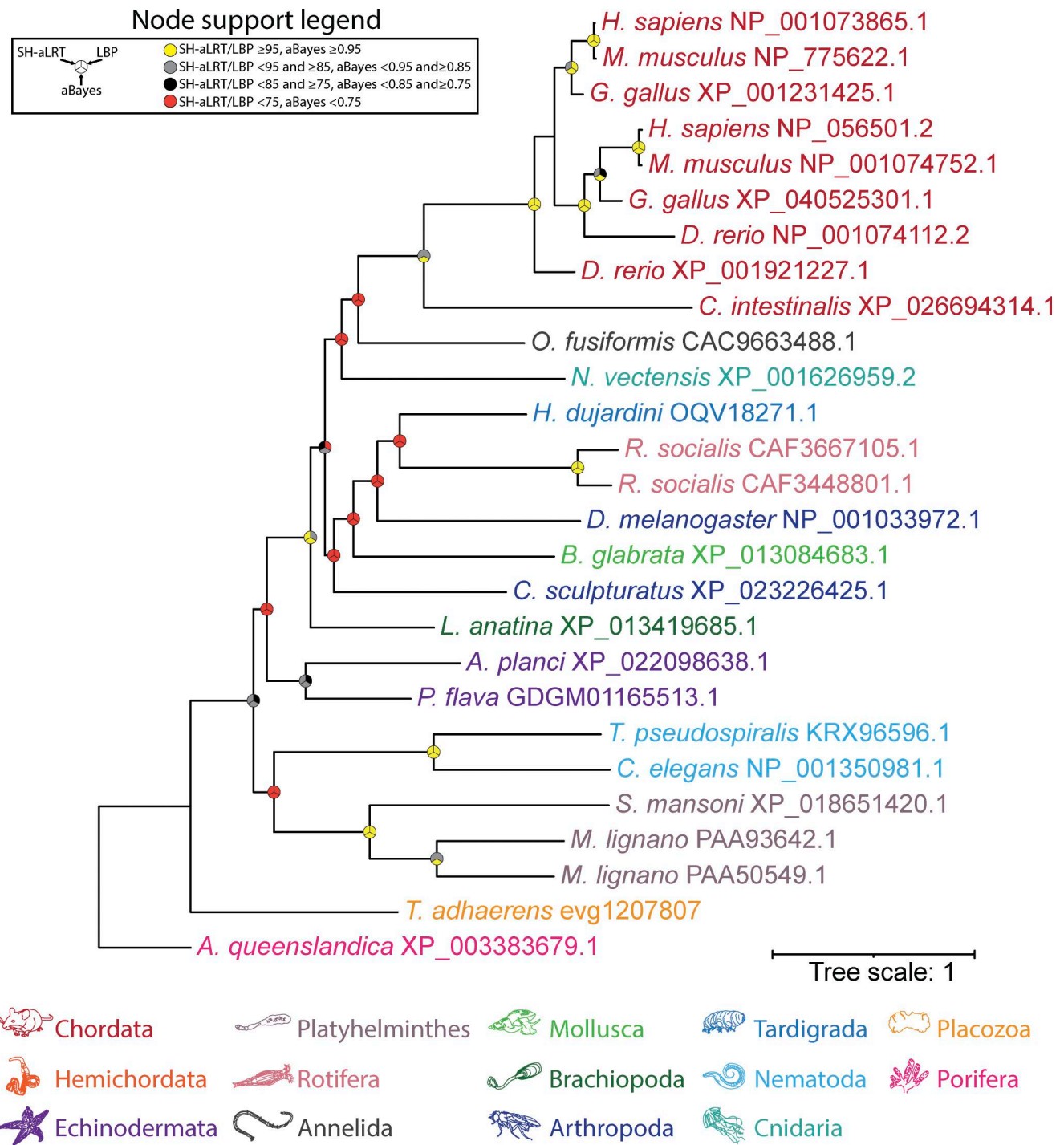

Figure S2.   **Maximum likelihood tree of manually identified FAM155 homologues from animals.** Node support values for three separate analyses, SH-aLRT, LBP, and aBayes, are depicted by circular symbols with colors reflecting ranges of values as indicated in the legend. SH-aLRT, Shimodaira–Hasegawa approximate likelihood ratio tests; aBayes, approximate Bayes tests; LBP, local bootstrap probability tests.

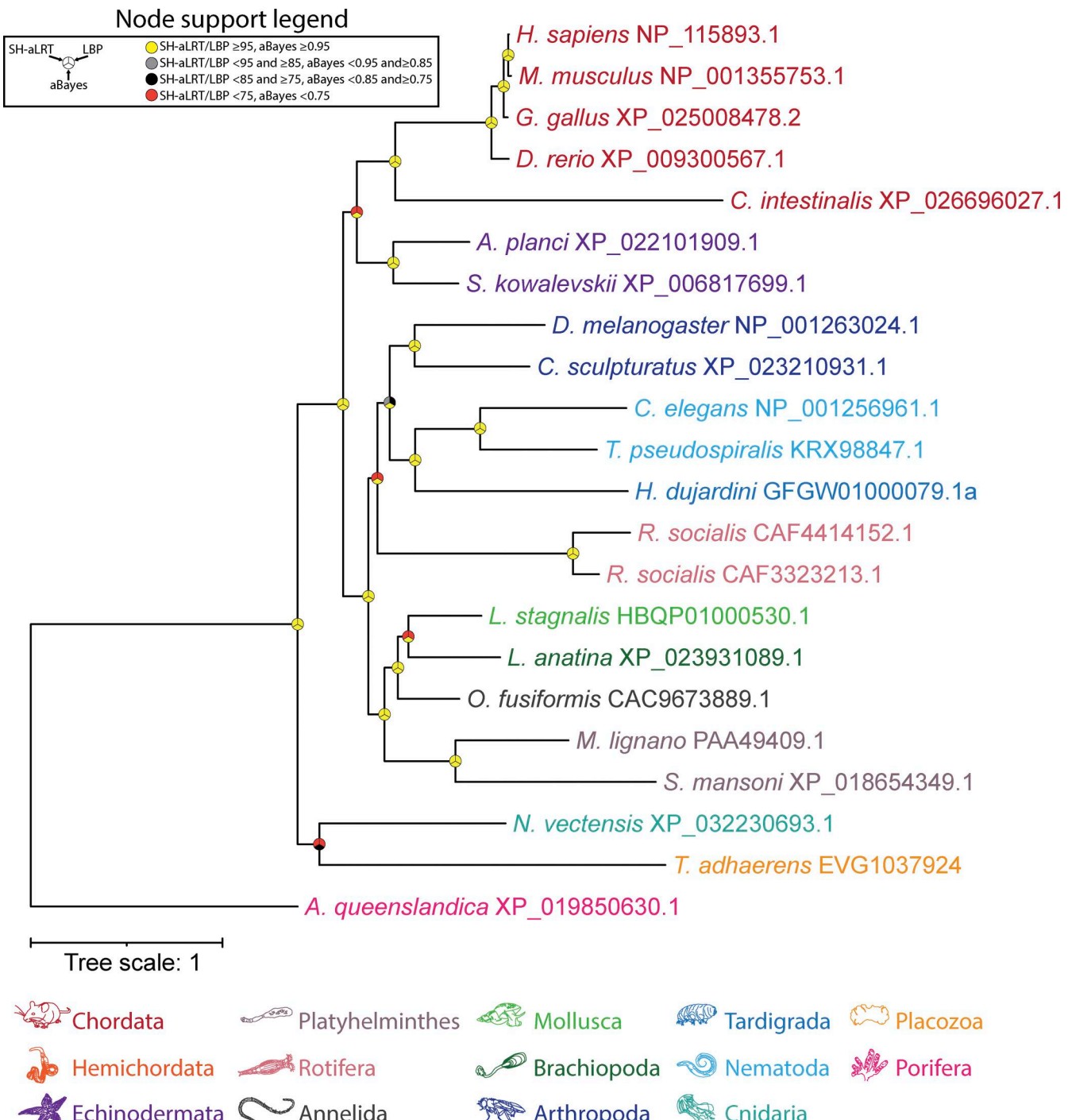

**Figure S3. Maximum likelihood tree of manually identified UNC79 homologues from animals.** Node support values for three separate analyses, SH-aLRT, LBP, and aBayes, are depicted by circular symbols with colors reflecting ranges of values as indicated in the legend. SH-aLRT, Shimodaira–Hasegawa approximate likelihood ratio tests; aBayes, approximate Bayes tests; LBP, local bootstrap probability tests.

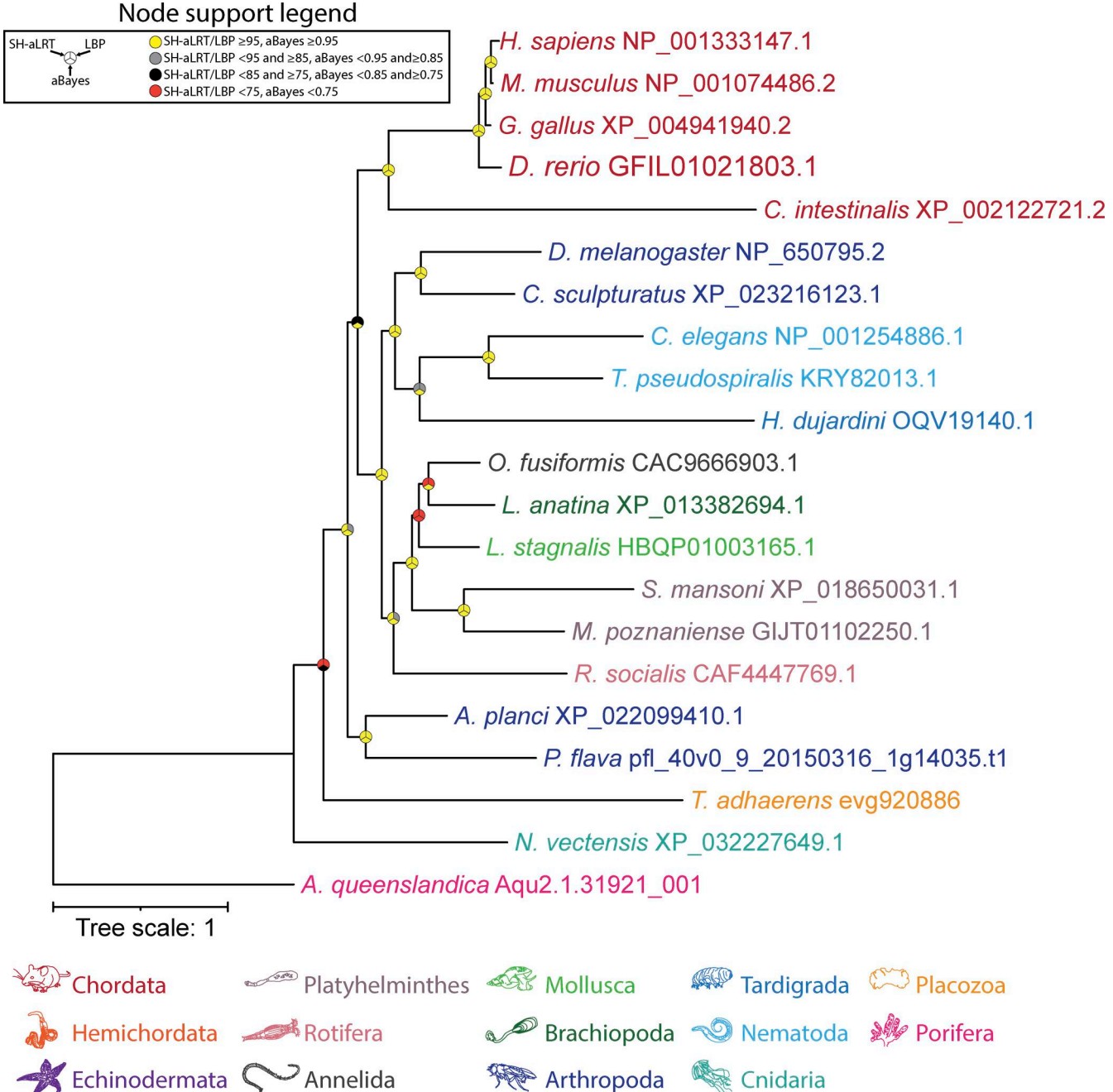

Figure S4. **Maximum likelihood tree of manually identified UNC80 homologues from animals.** Node support values for three separate analyses, SH-aLRT, LBP, and aBayes, are depicted by circular symbols with colors reflecting ranges of values as indicated in the legend. SH-aLRT, Shimodaira–Hasegawa approximate likelihood ratio tests; aBayes, approximate Bayes tests; LBP, local bootstrap probability tests.

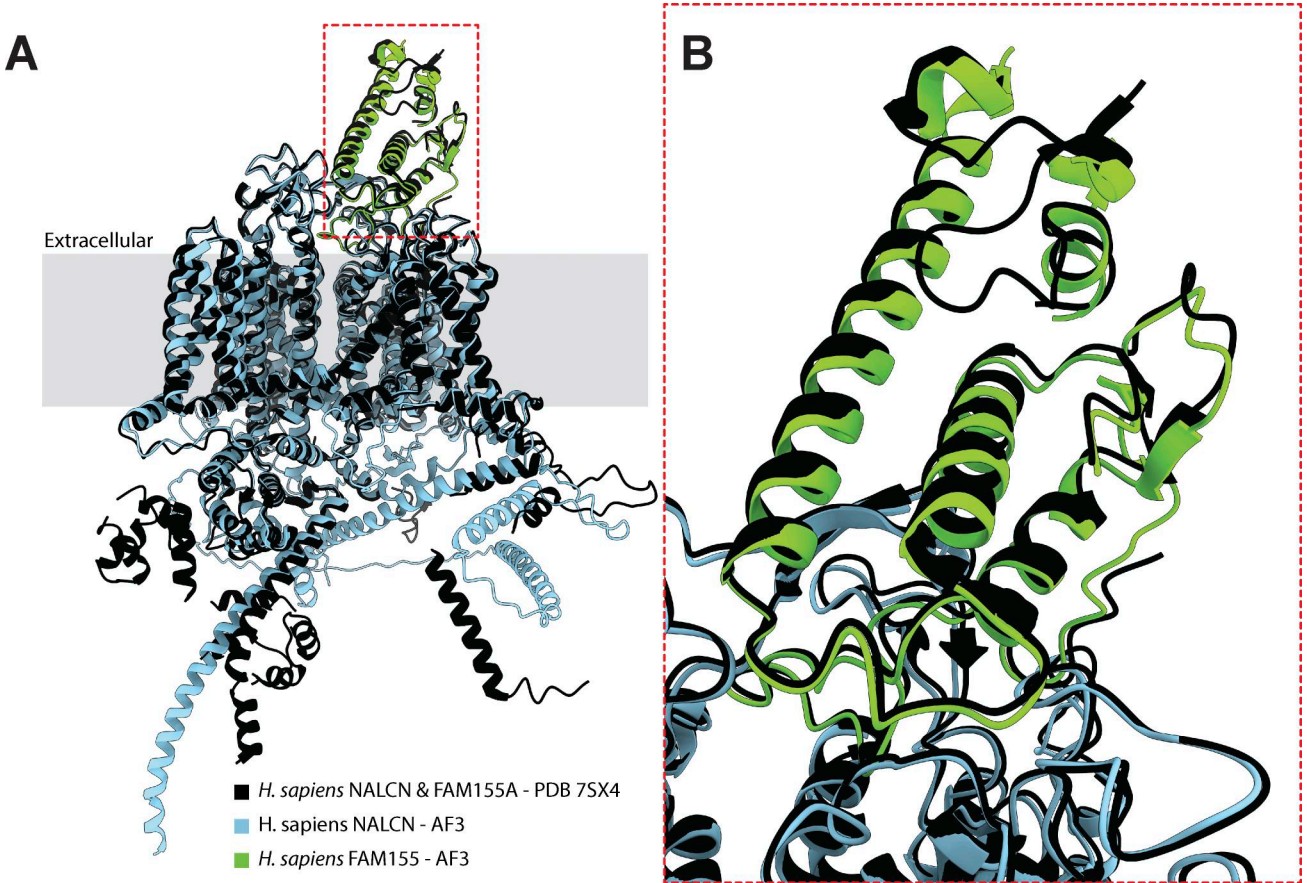

**A**

Extracellular

- ■ *H. sapiens* NALCN & FAM155A - PDB 7SX4
- ■ H. sapiens NALCN - AF3
- ■ *H. sapiens* FAM155 - AF3

**B**

Figure S5.  **Comparison of solved and predicted structures of the human NALCN-FAM155 dimer. (A)** Structural alignment of the solved structure of human NALCN and FAM155A (from PDB accession number 7SX4) and AlphaFold3-predicted structures of these proteins in complex with each other. **(B)** Close-up view of the region corresponding to the red dashed box in panel A. A version of these predicted structures, colored according to pLDDT confidence scores, is provided in Fig. S6 A.

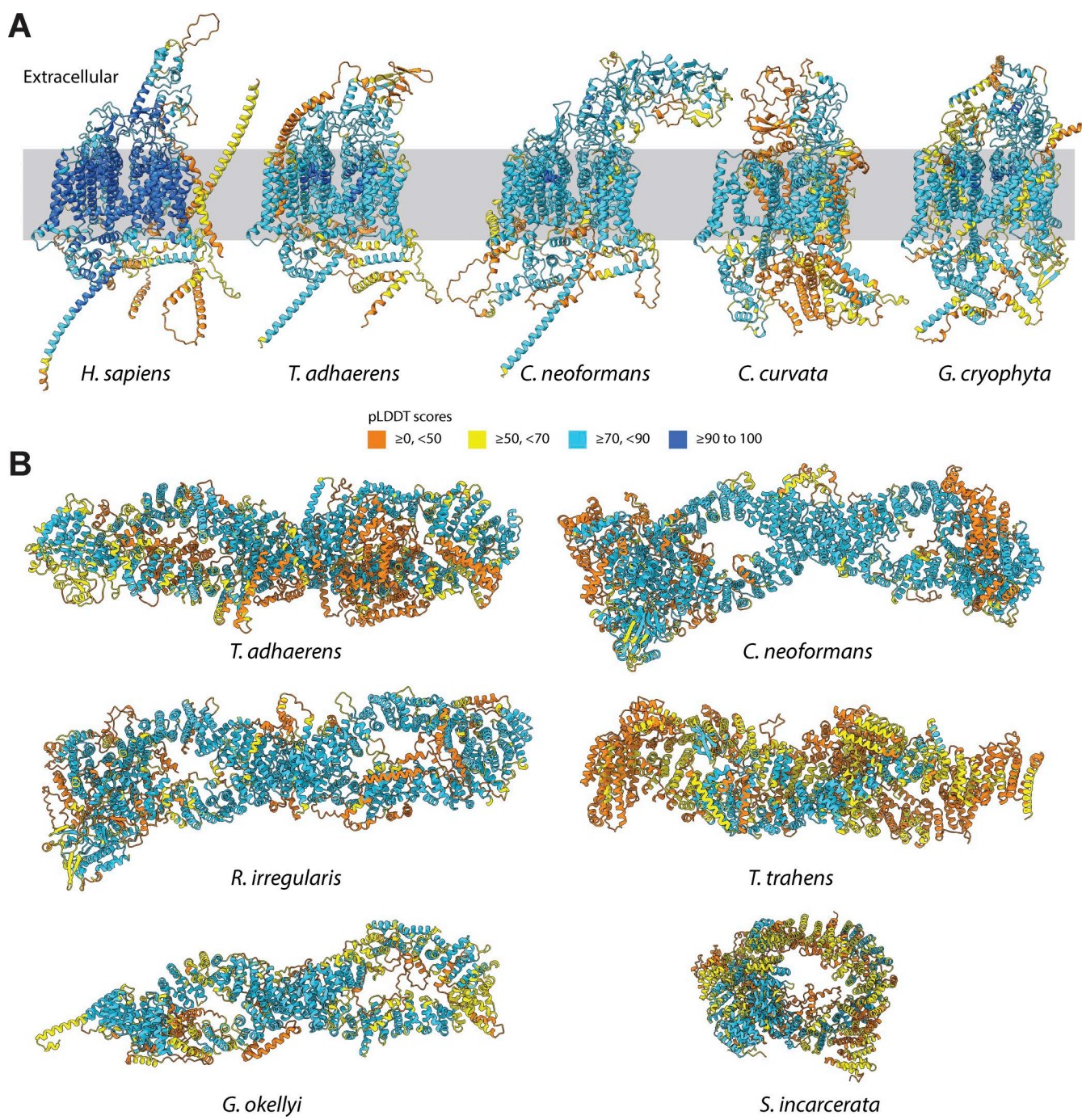

Figure S6. **Depictions of all presented AlphaFold predicted structures colored according to pLDDT confidence scores. (A)** Predicted structures of NALCN–FAM155 and Cch1-Mid1 dimers, colored according to AlphaFold pLDDT confidence scores. **(B)** Predicted structures of UNC79-UNC80 dimers, colored according to AlphaFold pLDDT confidence scores. The legend in the middle of the two panels indicates the range of pLDDT scores used and their corresponding colors.

**Provided online are 2 tables and 31 datasets. Table S1 provides details about the eukaryotic proteomes used in this study. Table S2 provides details about species, accession numbers, and AlphaFold confidence scores for sets of proteins analyzed in this study. Data S1 provides protein sequences and a global alignment of the UNC80 homologues from human and the fungal species R. delemar. Data S2 provides protein sequences of metazoan and fungal NALCN/Cch1 subunits used to generate a profile HMM model. Data S3 provides protein sequences of metazoan and fungal FAM155/Mid1 subunits used to generate a profile HMM model. Data S4 provides protein sequences of metazoan and fungal UNC80 subunits used to generate a profile HMM model. Data S5 provides protein**

sequences of fungal UNC80 subunits used to generate a profile HMM model. Data S6 provides protein sequences of metazoan and fungal UNC79 subunits used to generate a profile HMM model. Data S7 provides protein sequences of fungal UNC79 subunits used to generate a profile HMM model. Data S8 provides protein sequences of identified four domain channels, including NALCN and Cch1, from the set of eukaryotic proteomes. Data S9 provides protein sequences of identified Cch1 homologues from the set of fungal proteomes. Data S10 provides protein sequences of identified FAM155/Mid1 homologues from the set of eukaryotic proteomes. Data S11 provides protein sequences of identified Mid1 homologues from the set of fungal proteomes. Data S12 provides protein sequences of identified UNC80 homologues from the set of eukaryotic proteomes. Data S13 provides protein sequences of identified UNC80 homologues from the set of fungal proteomes. Data S14 provides sequences of identified UNC79 homologues from the set of eukaryotic proteomes. Data S15 provides sequences of identified UNC79 homologues from the set of fungal proteomes. Data S16 provides protein sequences of identified NALCN homologues from animals. Data S17 provides protein sequences of identified FAM155 homologues from animals. Data S18 provides protein sequences of identified UNC80 homologues from animals. Data S19 provides protein sequences of identified UNC79 homologues from animals. Data S20 provides the raw phylogenetic tree, in nexus format, of Fig. 2 A. Data S21 provides the raw phylogenetic tree, in nexus format, of Fig. 2 B. Data S22 provides the raw phylogenetic tree, in nexus format, of Fig. 3 B. Data S23 provides the raw phylogenetic tree, in nexus format, of Fig. 3 C. Data S24 provides the raw phylogenetic tree, in nexus format, of Fig. 5 B. Data S25 provides the raw phylogenetic tree, in nexus format, of Fig. 5 C. Data S26 provides the raw phylogenetic tree, in nexus format, of Fig. 6 B. Data S27 provides the raw phylogenetic tree, in nexus format, of Fig. 6 C. Data S28 provides the raw phylogenetic tree, in nexus format, of Fig. S1. Data S29 provides the raw phylogenetic tree, in nexus format, of Fig. S2. Data S30 provides the raw phylogenetic tree, in nexus format, of Fig. S3. Data S31 provides the raw phylogenetic tree, in nexus format, of Fig. S4.

