## [Peer Review File · The Journal of General Physiology]

NALCN/Cch1 channelosome subunits originated in early eukaryotes

Adriano Senatore, Tatiana Mayorova, Luis Yanez-Guerra, Wassim Elkhatib, Brian Bejoy, Philippe Lory, and Arnaud Monteil

Corresponding Author(s): Adriano Senatore, University of Toronto

Review Timeline:

Submission Date:	July 8, 2024
Editorial Decision:	August 20, 2024
Revision Received:	May 16, 2025
Editorial Decision:	June 24, 2025
Revision Received:	July 16, 2025
Editorial Decision:	July 22, 2025
Revision Received:	August 13, 2025

Editor: Crina Nimigean

Transaction Report:

DOI: <https://doi.org/10.1085/jgp.202413636>

August 20, 2024

Dr. Adriano Senatore
University of Toronto
Biology
3359 Mississauga Rd.
William G. Davis Building Rm. 3033
Mississauga, Ontario L5L 1C6
Canada

Re: 202413636

Dear Dr. Senatore,

Thank you for submitting your manuscript, entitled "Phylogenetic and sequence structure analysis of the NALCN/CCH1 channelosome complex provides new insights and uncovers an UNC-80 homologue in fungi." to JGP. Your manuscript has now been seen by 3 reviewers, whose comments are appended below. You will see that while the reviewers appreciated the careful and extensive analysis and the novel insights drawn, they have raised several concerns that should be addressed prior to further consideration of the manuscript at JGP. Most importantly, given the fact that the hypotheses advanced were not experimentally tested, the reviewers suggested and the editors agreed, that the manuscript should be considered as a "Viewpoint" rather than an "Article". With this in mind, please consider shortening and focusing your manuscript further to emphasize your novel findings by editing the results section according to the reviewers' recommendations, and reworking the section headings accordingly, and by placing the multiple sequence alignments in supplementary figures or an appendix. Furthermore, please make sure that you include a justification for the sequences you selected for analysis, as well as a presentation of the limitations of the study due to the small sample size.

We would be pleased to receive a suitably revised manuscript that addresses these concerns, which will be re-reviewed, most likely by some or all of the original referees. In addition, please do not hesitate to contact me (via the editorial office) if you feel that a discussion of the reviewers' and editors' comments would be helpful.

Please submit your revised manuscript via the link below along with a point-by-point letter that details your responses to the editors' and reviewers' comments, as well as a copy of the text with alterations highlighted (boldfaced or underlined). If the article is eventually accepted, it would include a 'revised date' as well as submitted and accepted dates. If we do not receive the revised manuscript within one year, we will regard the article as having been withdrawn. We would be willing to receive a revision of the manuscript at a later time, but the manuscript will then be treated as a new submission, with a new manuscript number.

Please pay particular attention to recent changes to our instructions to authors in the following sections: Data presentation, Blinding and randomization and Statistical analysis, under Materials and Methods, as shown here: <https://rupress.org/jgp/pages/submission-guidelines#prepare>. Re-review will be contingent on inclusion of the required information (including for data added during revision) and demonstration of the experimental reproducibility of the results. Also, to improve the reproducibility of published content, we have partnered with SciScore. Authors are prompted in eJP to copy and paste the Materials and Methods section of their manuscript for a SciScore assessment when submitting their revised manuscript. Authors are encouraged (not required) to further revise their Materials and Methods if the SciScore is below 4. More information can be found here: <https://rupress.org/jgp/pages/submission-guidelines#sciscore>

Please note, JGP now requires authors to submit Source Data used to generate figures containing gels and Western blots with all revised manuscripts (when applicable). This Source Data consists of fully uncropped and unprocessed images for each gel/blot displayed in the main and supplemental figures. If your paper includes cropped gel and/or blot images, please be sure to provide one Source Data file for each figure that contains gels and/or blots along with your revised manuscript files. File names for Source Data figures should be alphanumeric without any spaces or special characters (i.e., SourceDataF#, where F# refers to the associated main figure number or SourceDataFS# for those associated with Supplementary figures). The lanes of the gels/blots should be labeled as they are in the associated figure, the place where cropping was applied should be marked (with a box), and molecular weight/size standards should be labeled wherever possible. Source Data files will be made available to reviewers during evaluation of revised manuscripts and, if your paper is eventually published in JGP, the files will be directly linked to specific figures in the published article.

Source Data Figures should be provided as individual PDF files (one file per figure). Authors should endeavor to retain a minimum resolution of 300 dpi or pixels per inch. Please review our instructions for export from Photoshop, Illustrator, and PowerPoint here: <https://rupress.org/jgp/pages/submission-guidelines#revised>

When revising your manuscript, please be sure it is a double-spaced MS Word file and that it includes editable tables, if

appropriate.

Please submit your revised manuscript via this link:
Link Not Available

Thank you for the opportunity to consider your manuscript.

Sincerely,

Crina Nimigean, Ph.D.
On behalf of Journal of General Physiology

Journal of General Physiology's mission is to publish mechanistic and quantitative molecular and cellular physiology of the highest quality; to provide a best-in-class author experience; and to nurture future generations of independent researchers.

Reviewer #1 (Comments to the Authors):

Adriano Senatore and colleagues present their study on an updated phylogenetic analysis of NALCN, UNC-79, UNC-80, and FAM155. Observations regarding the evolution of the NALCN selectivity filter and voltage sensor are consistent with previous studies. NALCN is a channel responsible for sodium leak in cells and is embryonic lethal in homozygous knockout mice. Mutations of NALCN in humans result in intellectual, development, and locomotor disabilities.

This analysis comprises of 28 NALCN sequences, 21 UNC-79 sequences, 54 UNC-80 sequences, and 31 FAM155 sequences - which is roughly a doubling of NALCN sequence number from previous analyses (Liebeskind et al. Mol Biol Evol 2012) and bootstrap support for each branch is robust. Notable and novel to this study, the authors discover that fungi also have UNC-80 homologs but not UNC-79. These findings would be of interest to the broad scientific community and could drive new concepts in NALCN functional evolution and experimental efforts.

However, I would like the authors to elaborate on their approach for reproducibility. Perhaps key is how the sequence dataset was curated for NALCN, UNC-79, UNC-80, and FAM155. For a re-interrogation of the evolution of NALCN and its subunits I would have expected a larger sequence sampling. Is there little variance across the sequences (>95% identity)? A blast query of NALCN easily yields on the order of thousands of sequences. For databases outside of NCBI, there are approximately 300 sequenced fungal genomes (Araujo and Sampaio-Maia Adv Appl Microbiol 2018). I would imagine that UNC-79, UNC-80, and FAM155 likely have on the order of similar numbers. How were the specific sequences chosen? If the authors included more sequences in their analysis, support for their conclusions would be even more compelling. Relative to most modern phylogenetic studies, this is a rather small dataset.

Are the authors surprised that the best fit model is different for each alignment (NALCN, UNC-79, UNC-80, and FAM155)? Do they think this speaks towards the divergence of each family or the sequence sampling?

The authors describe the divergence of the sequences for the EF-hand like region for NALCN. For the EF-hand like region of NALCN channels could the authors provide an alignment with the canonical EF-hand and IQ domains of Nav and Cav channels?

Minor Comments:

In the cluster mapping, did the authors try any other substitution matrices other than BLOSUM62? Were there any differences?

Did the authors try any other algorithms than Phyre2 for predicting protein structures (alphafold, rosetta, or modeller for example)? Why was Phyre2 chosen?

In general, editing of the figures and legends would be helpful for clarity. For example, for Fig 10A and B, only the red trace and box are specifically described but not the blue traces. In Fig. 10D, it is difficult to see the highlighted substructures (what is UNC-79 NT is pointing to?, etc). Many of the alignments do not include any labeling of amino acid residue number.

Reviewer #2 (Comments to the Authors):

In this work by Senatore et al, the authors provide new insights into the phylogeny and sequence structure of NALCN/CCH1 channel complexes. The work highlights regions of high/low conservation in the various components of the complexes and uncovers a potential UNC80 analog in fungi, along with a region of possible UNC79/80 homology. The work is generally well done and presented logically (but see minor figure comments below). Not much is known about these

NALCN/CCH1 channel complexes, so the work will be of relevance to those in this small, but growing field. In particular the possibility of UNC80 homodimers and the notion of UNC79/80 representing ancient paralogs is intriguing, although both hypotheses would require validation by experimental approaches.

Major points:

- Overall, most sections in the results part read like a review. In other words, almost all results sections are quite lengthy heavy on the background. And although they interesting, it is hard to follow which parts exactly are novel. My recommendation would be to shorten each of the sections and have greater emphasis on what this work contributes. This could be done in form of short summaries at the end of each sections or similar, and would make the work more accessible to non-experts
- The authors mention a region of potential homology between UNC79 and UNC80 based on "weak but appreciable sequence conservation". But in the absence of any kind of benchmarking/scoring, this remains very vague. This is relevant in particular because it remains unclear how this situation differs from what is observed with e.g. importins etc. This needs to be clarified.
- In many places, the authors highlight the level of conservation of certain sequence stretches or specific residues. However, it remains unclear if the level of conservation translates into any degree of structural or functional relevance. In other words: is there evidence for any of the (many) highlighted conserved sequence stretches / side chains to play a structural or functional role? This might be apparent for some (e.g. NALCN patient variant R1092Q), but remains unclear for most. It would be helpful if the authors could either provide functional data to test predictions from their sequence analysis, or at least be clearer about the limitations of their analysis.

Minor points:

- The authors refer to large differences in the membrane topology among FAM155A and its relatives, but it remains unclear what this adds or what the implications are of this. Similarly, and in light of the conflicting reports regarding the subcellular localization of FAM155A/NLF-1 etc, could the authors speculate on potential species differences and possible downstream implications?
- Given that many figures deal with different proteins, it would be helpful for the reader to clearly indicate the protein / protein domain in question in each of the figures (not just in the legends)

Reviewer #3 (Comments to the Authors):

Peer Review | JGP Manuscript#: 202413636

Title: Phylogenetic and sequence structure analysis of the NALCN/CCH1 channelosome complex provides new insights and uncovers an UNC-80 homologue in fungi

Authors: Senatore et al.

SUMMARY

This manuscript by Senatore et al. delves into the evolution of the NALCN/CCH1 channelosome complex, as well as how the proteins in the complex have diverged over time.

The only novel finding in the present work is the identification of UNC-80 homologues in two Fungi, and the identified sequences are the result of a simple BLAST search. This limited novelty, combined with an extensive analysis of available sequences leading to numerous untested hypotheses does not constitute enough for publication as an original research "Article". Having said this, the presented analysis is extensive, and careful, and while insight in the absence of experiments is limited, publication in the special issue on "Molecular Evolution in the Membrane" as a "Hypothesis", "Perspective", "Analysis", or "Review" may be warranted.

COMMENTS

- Caption for Figure 1: The PDBID you provide is "7XS4" which refers to for "Crystal structure of URT1 in complex with AAAU RNA". I assume you meant to provide "7SX4: Human NALCN-FAM155A-UNC79-UNC80 channelosome with CaM bound, conformation 2/2".

- Caption for Figure 3: You say 1000 bootstraps were performed, but values seem to peak at 100 in all presented trees. Please clarify.

- "That the NALCN pore-forming subunits duplicated independently in *M. lignano*, *C. elegans*, and *A. queenslandica* is suggested by their respective monophyletic relationships on a protein phylogenetic tree (Figure 3)." Is this really evidence for independent parallel duplications? Other possibilities exist. Does this need to be stated?

- There are too many figures that focus on too many MSAs. I suggest condensing these, or making the alignments an Appendix or simply shared data.

- The "Results and Discussion" is literally a litany of hypotheses strung together from extensive and detailed analysis, and the "Conclusions" section reads more like "Future work". For this to be an original research "Article", I would argue that the "conclusions"/Future work should have been tested in the present work! A prime example being the hypothesis that UNC-80 can function as a homodimer, and thus eliminate the need for UNC-79 in some Fungi. If this can be published as a "Perspectives" or "Hypotheses" the section headings need to be made more appropriate.

We sincerely thank the reviewers for their time reviewing the manuscript and for their excellent and insightful comments. We hope this completely revised manuscript, with all new analyses and figures, and numerous new findings, warrants publication as a primary research article rather than a review.

Reviewer #1 (Comments to the Authors):

Adriano Senatore and colleagues present their study on an updated phylogenetic analysis of NALCN, UNC-79, UNC-80, and FAM155. Observations regarding the evolution of the NALCN selectivity filter and voltage sensor are consistent with previous studies. NALCN is a channel responsible for sodium leak in cells and is embryonic lethal in homozygous knockout mice. Mutations of NALCN in humans result in intellectual, development, and locomotor disabilities.

This analysis comprises of 28 NALCN sequences, 21 UNC-79 sequences, 54 UNC-80 sequences, and 31 FAM155 sequences - which is roughly a doubling of NALCN sequence number from previous analyses (Liebeskind et al. Mol Biol Evol 2012) and bootstrap support for each branch is robust. Notable and novel to this study, the authors discover that fungi also have UNC-80 homologs but not UNC-79. These findings would be of interest to the broad scientific community and could drive new concepts in NALCN functional evolution and experimental efforts.

However, I would like the authors to elaborate on their approach for reproducibility. Perhaps key is how the sequence dataset was curated for NALCN, UNC-79, UNC-80, and FAM155. For a re-interrogation of the evolution of NALCN and its subunits I would have expected a larger sequence sampling. Is there little variance across the sequences (>95% identity)? A blast query of NALCN easily yields on the order of thousands of sequences. For databases outside of NCBI, there are approximately 300 sequenced fungal genomes (Araujo and Sampaio-Maia Adv Appl Microbiol 2018). I would imagine that UNC-79, UNC-80, and FAM155 likely have on the order of similar numbers. How were the specific sequences chosen? If the authors included more sequences in their analysis, support for their conclusions would be even more compelling. Relative to most modern phylogenetic studies, this is a rather small dataset.

We agree with this and therefore conducted completely new phylogenetic analyses of all four NALCN channelosome subunits, using a more robust sequence identification strategy. Specifically, we extracted homologues from 184 high quality eukaryotic proteomes, and separately from a set of 256 fungal proteomes from the FungiDB database. Instead of using BLAST to identify homologues, which can fail to detect distant homology, we generated custom profile hidden Markov models of each subunit, and combined this with clustering,

phylogenetic analysis, and in some cases structural prediction to explore homology. This new analysis greatly expands our previous observations, revealing a much more ancestral origin of all four subunits.

Are the authors surprised that the best fit model is different for each alignment (NALCN, UNC-79, UNC-80, and FAM155)? Do they think this speaks towards the divergence of each family or the sequence sampling?

We agree that the different statistical models, which were selected by ModelFinder algorithm, likely reflect differences in sequence divergence rates among the NALCN subunits. Indeed, despite our significantly expanded analysis, different models were selected indicating differences in models in this analysis are not likely due to low sequence sampling.

The authors describe the divergence of the sequences for the EF-hand like region for NALCN. For the EF-hand like region of NALCN channels could the authors provide an alignment with the canonical EF-hand and IQ domains of Nav and Cav channels?

We decided to remove this and most other alignments from this version of the manuscript, focusing instead on the novel phylogenetic findings. This was done in response to concerns raised by reviewers 2 and 3 that the previous manuscript read more like a review or hypothesis than a research paper. We are considering revising our previous sequence analyses and alignments for a separate, brief review paper, after this work is published.

Minor Comments:

In the cluster mapping, did the authors try any other substitution matrices other than BLOSUM62? Were there any differences?

In this new analysis, we indeed tested different matrices for cluster mapping, as well as expect value cut-offs. We selected PAM30 for NALCN, given its moderate sequence homology with other four domain channels. For FAM155/Mid1, we used BLOSUM45, more appropriate for highly divergent sequences, and BLOSUM65 for UNC79 and UNC80, also for divergent sequences. We indicate the following in the materials and methods: "These various substitution matrices and expect value cutoffs were selected by comparing CLANS outputs from different parameters and selecting those that produced the best clustering."

Did the authors try any other algorithms than Phyre2 for predicting protein structures (alphafold, rosetta, or modeller for example)? Why was Phyre2 chosen?

In this new analysis we only used AlphaFold3 given its capacity to predict protein complexes.

In general, editing of the figures and legends would be helpful for clarity. For example, for Fig 10A and B, only the red trace and box are specifically described but not the blue traces. In Fig. 10D, it is difficult to see the highlighted substructures (what is UNC-79 NT is pointing to?, etc). Many of the alignments do not include any labeling of amino acid residue number.

All the figures in this revised manuscript are new and based on new analyses. In figure 7, where we include predicted UNC79-UN80 dimeric structures, we added the following to the figure legend: “For all panels, the labels NT and CT denote the rough location of the N- and C-termini of UNC79 and UNC80 proteins, respectively, along a horizontal plane.”

Reviewer #2 (Comments to the Authors):

In this work by Senatore et al, the authors provide new insights into the phylogeny and sequence structure of NALCN/CCH1 channel complexes. The work highlights regions of high/low conservation in the various components of the complexes and uncovers a potential UNC80 analog in fungi, along with a region of possible UNC79/80 homology.

The work is generally well done and presented logically (but see minor figure comments below). Not much is known about these NALCN/CCH1 channel complexes, so the work will be of relevance to those in this small, but growing field. In particular the possibility of UNC80 homodimers and the notion of UNC79/80 representing ancient paralogs is intriguing, although both hypotheses would require validation by experimental approaches.

Major points:

- Overall, most sections in the results part read like a review. In other words, almost all results sections are quite lengthy heavy on the background. And although they interesting, it is hard to follow which parts exactly are novel. My recommendation would be to shorten each of the sections and have greater emphasis on what this work contributes. This could be done in form of short summaries at the end of each sections or similar, and would make the work more accessible to non-experts

We recognize that the first version of this manuscript was more like a review, and it was unclear what the novel findings were. In this version, as noted above to reviewer 1, we present completely new analyses that are much more robust, and appropriate for publication as primary research. Since all previous figures were removed, and new ones generated, we completely rewrote the paper to focus exclusively on the research findings.

- The authors mention a region of potential homology between UNC79 and UNC80 based on "weak but appreciable sequence conservation". But in the absence of any kind of benchmarking/scoring, this remains very vague. This is relevant in particular because it remains unclear how this situation differs from what is observed with e.g. importins etc. This needs to be clarified.

This aspect of the previous manuscript was also a source of concern for reviewer 3. We therefore removed this and may include it in a separate review paper.

- In many places, the authors highlight the level of conservation of certain sequence stretches or specific residues. However, it remains unclear if the level of conservation translates into any degree of structural or functional relevance. In other words: is there evidence for any of the (many) highlighted conserved sequence stretches / side chains to play a structural or functional role? This might be apparent for some (e.g. NALCN patient

variant R1092Q), but remains unclear for most. It would be helpful if the authors could either provide functional data to test predictions from their sequence analysis, or at least be clearer about the limitations of their analysis.

This new version of the manuscript does not include the previous alignments and references to sequence conservation therein.

Minor points:

- The authors refer to large differences in the membrane topology among FAM155A and its relatives, but it remains unclear what this adds or what the implications are of this. Similarly, and in light of the conflicting reports regarding the subcellular localization of FAM155A/NLF-1 etc, could the authors speculate on potential species differences and possible downstream implications?

We have repeated the analysis of FAM155/Mid1 proteins, focusing on the discovery of homologues in cryptists, and corroborating the existence of this subunit in fungi. We also include in an alignment the partial sequence of a FAM155/Mid1 homologue from an apusomonad species that was previously reported (DOI: 10.3389/fnmol.2014.00015). In this new analysis, we used AlphaFold predicted structures to show that the α 1 to α 3 helices are conserved in metazoan, fungal and cryptist FAM155/Mid1 homologues. Unlike our previous analysis, we do not include predictions of membrane helices nor signal peptides hence a discussion about membrane topology is not warranted, and perhaps more appropriate for a follow up review paper.

- Given that many figures deal with different proteins, it would be helpful for the reader to clearly indicate the protein / protein domain in question in each of the figures (not just in the legends)

Our new streamlined analysis does not include the previous alignments, and we hope it is evident in the new figures and figure legends which subunit is being referred to.

Reviewer #3 (Comments to the Authors):

Peer Review | JGP Manuscript#: 202413636

Title: Phylogenetic and sequence structure analysis of the NALCN/CCH1 channelosome complex provides new insights and uncovers an UNC-80 homologue in fungi

Authors: Senatore et al.

SUMMARY

This manuscript by Senatore et al. delves into the evolution of the NALCN/CCH1 channelosome complex, as well as how the proteins in the complex have diverged over time.

The only novel finding in the present work is the identification of UNC-80 homologues in two Fungi, and the identified sequences are the result of a simple BLAST search. This limited novelty, combined with an extensive analysis of available sequences leading to numerous untested hypotheses does not constitute enough for publication as an original research "Article". Having said this, the presented analysis is extensive, and careful, and while insight in the absence of experiments is limited, publication in the special issue on "Molecular Evolution in the Membrane" as a "Hypothesis", "Perspective", "Analysis", or "Review" may be warranted.

As noted above, we agree with this assessment and therefore conducted a more comprehensive and rigorous analysis that we feel meets the standard for publication as a primary research article. All the figures and analyses are new (including the supplementary phylogenies of metazoan subunits which were re-run with three separate node support approximation methods). This new analysis reveals a much more ancestral origin for NALCN subunits, which we think will be useful knowledge for researchers in the separate fields on NALCN and CCh1 channel research.

COMMENTS

- Caption for Figure 1: The PDBID you provide is "7XS4" which refers to for "Crystal structure of URT1 in complex with AAAU RNA". I assume you meant to provide "7SX4: Human NALCN-FAM155A-UNC79-UNC80 channelosome with CaM bound, conformation 2/2".

Thank you for pointing out this mistake. The new manuscript has the correct PDB number indicated.

- Caption for Figure 3: You say 1000 bootstraps were performed, but values seem to peak at 100 in all presented trees. Please clarify.

All bootstrap values are presented as percentages. The following is indicated in the methods section:

“For all trees, node support was estimated via 1000 replicate Shimodaira-Hasegawa approximate likelihood ratio tests (SH-aLRT)⁴⁸, approximate Bayes tests (aBayes)⁴⁹, and fast local bootstrap probability tests (LBP)⁵⁰. All support values are listed as percentages.”

- "That the NALCN pore-forming subunits duplicated independently in *M. lignano*, *C. elegans*, and *A. queenslandica* is suggested by their respective monophyletic relationships on a protein phylogenetic tree (Figure 3)." Is this really evidence for independent parallel duplications? Other possibilities exist. Does this need to be stated?

We have refrained from using the word “independent” when referring to our phylogenetic trees.

- There are too many figures that focus on too many MSAs. I suggest condensing these, or making the alignments an Appendix or simply shared data.

These have been removed.

- The "Results and Discussion" is literally a litany of hypotheses strung together from extensive and detailed analysis, and the "Conclusions" section reads more like "Future work". For this to be an original research "Article", I would argue that the "conclusions"/Future work should have been tested in the present work! A prime example being the hypothesis that UNC-80 can function as a homodimer, and thus eliminate the need for UNC-79 in some Fungi. If this can be published as a "Perspectives" or "Hypotheses" the section headings need to be made more appropriate.

The results and discussion in this version of the manuscript were completely rewritten, focusing primarily on our discovery of non-metazoan subunits.

June 24, 2025

Dr. Adriano Senatore
University of Toronto
Biology
3359 Mississauga Rd.
William G. Davis Building Rm. 3033
Mississauga, Ontario L5L 1C6
Canada

Re: 202413636R1

Dear Dr. Senatore,

Thank you for submitting your manuscript, entitled "NALCN/Cch1 channelosome subunits originated in early eukaryotes and a complete set is conserved in animals, fungi, and apusomonads" to JGP. Your manuscript has now been seen by the same 3 reviewers, whose comments are appended below. You will see that the reviewers were enthusiastic about the revisions and the potential impact of the revised manuscript and raised only minor concerns that should nevertheless be addressed prior to further consideration of the manuscript at JGP. Specifically, please consider a different title. Our recommendation is: "Remote origin of NALCN/Ch1 channelosome subunits and their conservation in present taxa". In addition, it would be helpful to provide the degree of confidence of the alphafold structures presented and also provide a short discussion paragraph about the level of conservation of the interfaces between the channel and the auxiliary subunits modeled compared to those experimentally determined.

We hope that you will be able to submit a revised manuscript that addresses these points, which we believe will pose no problems, and which may be re-reviewed. In addition, please do not hesitate to contact me (via the editorial office) if you feel that a discussion of the reviewers' and editors' comments would be helpful.

Please submit your revised manuscript via the link below, along with a point-by-point letter that details your response to the reviewers' and editors' comments, as well as a copy of the text with alterations highlighted (boldfaced or underlined). If the article is eventually accepted, it would include a 'revised date' as well as submitted and accepted dates. If we do not receive the revised manuscript within one year, we will regard the article as having been withdrawn. We would be willing to receive a revision of the manuscript at a later time, but the manuscript will then be treated as a new submission, with a new manuscript number.

Please pay particular attention to recent changes to our instructions to authors in the following sections: Data presentation, Blinding and randomization and Statistical analysis, under Materials and Methods, as shown here: <https://rupress.org/jgp/pages/submission-guidelines#prepare>. Re-review will be contingent on inclusion of the required information (including for data added during revision) and demonstration of the experimental reproducibility of the results. Also, To improve the reproducibility of published content, we have partnered with SciScore. Authors are prompted in eJP to copy and paste the Materials and Methods section of their manuscript for a SciScore assessment when submitting their revised manuscript. Authors are encouraged (not required) to further revise their Materials and Methods if the SciScore is below 4. More information can be found here: <https://rupress.org/jgp/pages/submission-guidelines#sciscore>.

Please note, JGP now requires authors to submit Source Data used to generate figures containing gels and Western blots with all revised manuscripts (when applicable). This Source Data consists of fully uncropped and unprocessed images for each gel/blot displayed in the main and supplemental figures. If your paper includes cropped gel and/or blot images, please be sure to provide one Source Data file for each figure that contains gels and/or blots along with your revised manuscript files. File names for Source Data figures should be alphanumeric without any spaces or special characters (i.e., SourceDataF#, where F# refers to the associated main figure number or SourceDataFS# for those associated with Supplementary figures). The lanes of the gels/blots should be labeled as they are in the associated figure, the place where cropping was applied should be marked (with a box), and molecular weight/size standards should be labeled wherever possible. Source Data files will be made available to reviewers during evaluation of revised manuscripts and, if your paper is eventually published in JGP, the files will be directly linked to specific figures in the published article.

Source Data Figures should be provided as individual PDF files (one file per figure). Authors should endeavor to retain a minimum resolution of 300 dpi or pixels per inch. Please review our instructions for export from Photoshop, Illustrator, and PowerPoint here: <https://rupress.org/jgp/pages/submission-guidelines#revised>

Whilst you are revising your manuscript, we ask that you consider whether you have any artwork that might be suitable for the cover of JGP. Microscopy images are particularly good for cover artwork, but other types of image can be very effective, so we

encourage you to be creative. Please don't restrict yourself to images from the paper; an image that is relevant to the work described would be just as suitable. Images should be a minimum resolution of 300 dpi. To see recent examples, visit the following page and click on 'Show covers? Yes': <https://jgp.rupress.org/content/by/year>

Thank you for submitting your interesting research to JGP.

Please submit your revised manuscript, and any associated files, via this link:
Link Not Available

Sincerely,

Crina Nimigean, Ph.D.
On behalf of Journal of General Physiology

Journal of General Physiology's mission is to publish mechanistic and quantitative molecular and cellular physiology of the highest quality; to provide a best-in-class author experience; and to nurture future generations of independent researchers.

Reviewer #1 (Comments to the Authors):

The revised manuscript from Adriano Senatore and colleagues is much improved and I appreciate the details in sequence curation and analysis. The authors have addressed my concerns appropriately.

Minor points:

Page 4, I found this sentence confusing - I suggest removing the parenthesized i.e.:
Metazoan CaV1 and CaV2 channels, CaV3 channels, and NaV channels fall within three distant and separate clades from NALCN/Cch1 channels and indeed all clade A channels (i.e., clades B, C, and D respectively) (Figure 2A), with each clade also containing other sets of channels from microbial eukaryotic taxa including Choanoflagellata, Chloroplastida, Cryptista, and Apusomonadida.

I would like the authors to quantitatively or qualitatively define what constitutes best clustering. Is this biased towards a priori knowledge? Methods:

These various substitution matrices and expect value cutoffs were selected by comparing CLANS outputs from different parameters and selecting those that produced the best clustering.

For legends of Fig 1, 3, 5, and 6, I would suggest to the authors to revise their descriptions of the CLANS panels to simply state that they are cluster maps. I recognize the algorithm compares against all sequence similarities, but I found the all vs all cluster map descriptors initially obscure.

Reviewer #2 (Comments to the Authors):

The authors of this study use bioinformatics and other computational approaches to identify essential subunits of the NALCN/Cch1 channelosome in a wide set of animals, as well as fungi and apusomonads. Given the sparsity of information on these subunits, the work is potentially important and will be of relevance to those working on NALCN, and possibly also the related Nav and Cav channels. The work is overall well-presented and reasoned, although many of the arguments would be considerably strengthened by experimental evidence. While I realise this may be beyond the scope of the present work, the authors should be more transparent about this shortcoming in their conclusions.

Additionally, there are a number of other issues that should be addressed:

- Title: the title should be a more nuanced reflection of the results. The statement that "a complete set is conserved in animals, fungi and apusomonads" is a bit misleading because in none of the taxonomic groupings do all of the considered species/lineages seem to have a complete set (Fig 8).
- On a related note: could the authors elaborate on the following statement: "In contrast, and as noted, UNC79 and UNC80 are less prevalent in fungi, suggesting Cch1/Mid1 can function independently from UNC79/UNC80, permitting the loss of the latter subunits in select fungal lineages.". What evidence makes them suggest that Cch1/Mid1 can function without the UNC proteins? Could it be that rather other proteins substitute for the UNC proteins? Or that Cch1/Mid1 are present in the genome, but not expressed or not functionally relevant? In other words: the work would benefit from a more nuanced (but not necessarily extensive) discussion on cases where the UNC proteins appear to be missing
- On a couple of occasions, FAM155 is referred to as an extracellular subunit. This is somewhat misleading, as the extracellular

domain of FAM155 is predicted to be flanked by both an N- and a C-terminal transmembrane helix.

- Related to the above, it is not apparent if any of the AF predictions shown in Fig 4A would produce steric clashes. Also, it might be helpful for the reader to be presented with heat map structures that outline the confidence with which AF predicts the assemblies shown in Fig 4 A/B (at least in the supplement).

Similarly, this should be done for the UNC79/80 assemblies shown in Fig 7. Otherwise, these depictions might appear overly confident or optimistic.

- In the discussion of the obvious molecular differences in e.g. animals, the authors focus primarily on the selectivity filter (SF). While this is interesting, I see two issues with it: first, it is not clear why the authors refer to the SF as "hypervariable" when really there are only two versions of it, i.e. EEKE and EEEE (plus, it remains unknown what the functional implications are of this difference). Second, and more importantly: Given that the novelty of the present work lies mostly in the identification of subunits other than the pore-forming subunit, it would be more appropriate to discuss potential implications of differences in the interaction between NALCN/Cch1 and FAM155/Mid1 or the UNC proteins. This is especially important given that the channel-UNC interfaces are crucial to the function of the channelosome. In other words: the discussion would benefit from more focus on this aspect, rather than the SF, i.e. are the channel-subunit interfaces conserved? can they be predicted?

Reviewer #3 (Comments to the Authors):

Peer Review | JGP Manuscript#: 202413636R1

Title: NALCN/Cch1 channelosome subunits originated in early eukaryotes and a complete set is conserved in animals, fungi, and apusomonads

The revised manuscript is much improved, easier to read, and the analysis and conclusions are more focussed.

MINOR COMMENTS

1. Page 6 of the PDF: '...predicted the structure of the human NALCN-FAM155A complex and structurally aligned it with its corresponding solved structure7, RECEALING highly overlapping structures with a root mean square alignment deviation score of 0.784 angstroms (Figure S3).' Correct the spelling mistake (REVEALING).

1. Line 1 of the Discussion: 'Our phylogenetic analysis of four domain channels identified using a custom HMM profile trained on NALCN and Cch1 protein sequences...' I'm not sure that it is appropriate to say the the HMM profile was 'trained on' a given set of sequences. Perhaps a better way of saying it would be that 'it was built from'? In the age of 'machine learning' the term 'trained on' means something very specific.

We would like to thank the reviewers for their time reading the revised manuscript, and for their valuable insights. We hope to have addressed all their concerns in this revised manuscript.

Reviewer #1 (Comments to the Authors):

The revised manuscript from Adriano Senatore and colleagues is much improved and I appreciate the details in sequence curation and analysis. The authors have addressed my concerns appropriately.

Minor points:

Page 4, I found this sentence confusing - I suggest removing the parenthesized i.e.:

Metazoan Ca_v1 and Ca_v2 channels, Ca_v3 channels, and Na_v channels fall within three distant and separate clades from NALCN/Cch1 channels and indeed all clade A channels (i.e., clades B, C, and D respectively) (Figure 2A), with each clade also containing other sets of channels from microbial eukaryotic taxa including Choanoflagellata, Chloroplastida, Cryptista, and Apusomonadida.

We agree that this sentence was not clear. We have therefore changed it to the following (lines 108 to 113):

“Metazoan Ca_v1 and Ca_v2 channels, Ca_v3 channels, and Na_v channels fall within three distant and separate clades respectively (i.e., clades B, C, and D), distant from NALCN/Cch1 channels and indeed all clade A channels (Figure 2A). All these three clades also contain other sets of channels from microbial eukaryotic taxa including Choanoflagellata, Chloroplastida, Cryptista, and Apusomonadida.”

I would like the authors to quantitatively or qualitatively define what constitutes best clustering. Is this biased towards a priori knowledge?

Methods:

These various substitution matrices and expect value cutoffs were selected by comparing CLANS outputs from different parameters and selecting those that produced the best clustering.

We realize that our description of how we chose parameters for the CLANS analysis of the different NALCN subunits was inadequate. We therefore revised the materials and methods to provide a clearer rationale as to how the different sets of parameters were selected (lines 457 to 464):

“The noted substitution matrices were chosen to reflect differences in sequence conservation among subunit homologues (*i.e.*, PAM30 for the four-domain channels with their highly conserved transmembrane helices, and BLOSUM45 and BLOSUM65 for FAM155 and UNC-79/UNC-80, respectively, due to their more divergent nature). Expect values were selected to ensure that all included sequences were linked to at least two other nodes.”

For legends of Fig 1, 3, 5, and 6, I would suggest to the authors to revise their descriptions of the CLANS panels to simply state that they are cluster maps. I recognize the algorithm compares against all sequence similarities, but I found the all vs all cluster map descriptors initially obscure.

We have changed these to indicate cluster maps as suggested.

Reviewer #2 (Comments to the Authors):

The authors of this study use bioinformatics and other computational approaches to identify essential subunits of the NALCN/Cch1 channelosome in a wide set of animals, as well as fungi and apusomonads. Given the sparsity of information on these subunits, the work is potentially important and will be of relevance to those working on NALCN, and possibly also the related Nav and Cav channels. The work is overall well-presented and reasoned, although many of the arguments would be considerably strengthened by experimental evidence. While I

realise this may be beyond the scope of the present work, the authors should be more transparent about this shortcoming in their conclusions.

Thank you for this suggestion. In response, we have revised and added text to the discussion to stress the importance for future functional studies of NALCN/Cch1 channelosome subunits outside of mammals (lines 341 to 360):

“Interestingly, the complete co-occurrence of all four NALCN subunits within animals suggests they have an obligate functional relationship within each other, which is consistent with the various genotype-phenotype studies that have been done in nematode worms, fruit flies, and mice (reviewed in³⁻⁵). In contrast, many lineages of fungi possess Cch1 and Mid1 but lack UNC79 and UNC80 (Figure 8). Extensive work in the brewer’s yeast *Saccharomyces cerevisiae* (Saccharomycotina) has revealed that Cch1 and Mid1 interact and function together^{10,29,30}, and the absence of UNC79 and UNC80 in this species indicates these two subunits can operate without the large cytoplasmic subunits, at least in some fungi. Furthermore, Cch1 and Mid1 from the basidiomycete fungus *C. neoformans* were reported to form functional Ca²⁺ channels *in vitro*^{20,31}, although this species possesses UNC79 and UNC80 (Figure 7), contrasting recent work on mammalian channels expressed *in vitro* where reconstitution of functional currents required co-expression of all four subunits^{32,33}. These observations raise interesting questions about the conserved or divergent functions of UNC79, UNC80, and indeed all NALCN/CCh1 subunits, within different organismal lineages. Additional questions surround the conservation of molecular contacts between subunits that are required for complex assembly. Finding answers to these questions will require functional and structural characterization of these channel complexes from an expanded set of eukaryotic organisms, including invertebrate animals^{34,35}. Such comparative work, and the establishment of a diversity of organisms for studying NALCN/Cch1 function, has the potential to uncover important insights into the molecular physiology of these channels and their subunits within a broad eukaryotic context.”

Additionally, there are a number of other issues that should be addressed:

- Title: the title should be a more nuanced reflection of the results.

The statement that "a complete set is conserved in animals, fungi and apusomonads" is a bit misleading because in none of the taxonomic groupings do all of the considered species/lineages seem to have a complete set (Fig 8).

We have changed to title to prevent such a misunderstanding:

“NALCN/Cch1 channelosome subunits originated in early eukaryotes and a full complement is found in select animals, fungi, and apusomonads”

- On a related note: could the authors elaborate on the following statement: "In contrast, and as noted, UNC79 and UNC80 are less prevalent in fungi, suggesting Cch1/Mid1 can function independently from UNC79/UNC80, permitting the loss of the latter subunits in select fungal lineages.". What evidence makes them suggest that Cch1/Mid1 can function without the UNC proteins? Could it be that rather other proteins substitute for the UNC proteins? Or that Cch1/Mid1 are present in the genome, but not expressed or not functionally relevant? In other words: the work would benefit from a more nuanced (but not necessarily extensive) discussion on cases where the UNC proteins appear to be missing

We hope that our amended discussion included above adequately clarifies that Cch1 and Mid1 are fully operational in fungal species that lack UNC79 and UNC80 (e.g., *S. cerevisiae*), and that even the homologues from *C. neoformans*, which possesses UNC79 and UNC80, are reported to form functional channels in vitro without the two cytoplasmic subunits.

- On a couple of occasions, FAM155 is referred to as an extracellular subunit. This is somewhat misleading, as the extracellular domain of FAM155 is predicted to be flanked by both an N- and a C-terminal transmembrane helix.

The reviewer raised an important point that many questions remain about the subcellular localization and membrane topology of the FAM155 subunit. Thus, we have removed the word “extracellular” throughout the manuscript when describing FAM155.

- Related to the above, it is not apparent if any of the AF predictions shown in Fig 4A would produce steric clashes. Also, it might be helpful for the reader to be presented with heat map structures that outline the confidence with which AF predicts the assemblies shown in Fig 4 A/B (at least in the supplement).

Since these are predicted structures, we avoided discussing specific residues within predicted protein interfaces that might contribute to attractions or clashes. Instead, we suggest that the provided ipTM values (Supplementary Table 2) provide necessary information about the confidence of the predicted dimer interfaces. However, we agree with the reviewer that a supplementary figure of the predicted structures, colored according to AlphaFold confidence values (pLDDT), is appropriate. Thus, we created a such a figure, Figure S4, with all predicted structures shown in Figures 4A/B and 7, colored according to AlphaFold pLDDT scores:

Figure S4. A) Predicted structures of NALCN-FAM155 and Cch1-Mid1 dimers colored according to AlphaFold pLDDT confidence scores. **B)** Predicted structures of UNC79-UNC80 dimers, colored according to AlphaFold pLDDT confidence scores. The legend in the middle of the two panels indicates the range of pLDDT scores used and their corresponding colors.

Similarly, this should be done for the UNC79/80 assemblies shown in Fig 7. Otherwise, these depictions might appear overly confident or optimistic.

We have done this. Please see our comment and new figure above.

- In the discussion of the obvious molecular differences in e.g. animals, the authors focus primarily on the selectivity filter (SF). While this is interesting, I see two issues with it: first, it is not clear why the authors refer to the SF as "hypervariable" when really there are only two versions of it, i.e. EEKE and EEEE (plus, it remains unknown what the functional implications are of this difference). Second, and more importantly: Given that the novelty of the present work lies mostly in the identification of subunits other than the pore-forming subunit, it would be more appropriate to discuss potential implications of differences in the interaction between NALCN/Cch1 and FAM155/Mid1 or the UNC proteins. This is especially important given that the channel-UNC interfaces are crucial to the function of the channelosome. In other words: the discussion would benefit from more focus on this aspect, rather than the SF, i.e. are the channel-subunit interfaces conserved? can they be predicted?

We agree that this topic is tangential to the work presented in this manuscript. Thus, we have removed all text describing the selectivity filter of NALCN channels. With respect to interfaces between subunits, like our comment above, we feel it best not to discuss specific residues involved in complexing based only on predicted structures. Nonetheless, we agree that this will be important to address experimentally going forward. Hence, we added to and amended our discussion as follows (lines 352 to 360):

“These observations raise interesting questions about the conserved or divergent functions of UNC79, UNC80, and indeed all NALCN/CCh1 subunits, within different organismal lineages. Additional questions surround the conservation of molecular contacts between subunits that are required for complex assembly. Finding answers to these questions will require functional

and structural characterization of these channel complexes from an expanded set of eukaryotic organisms, including invertebrate animals^{34,35}. Such comparative work, and the establishment of a diversity of organisms for studying NALCN/Cch1 function, has the potential to uncover important insights into the molecular physiology of these channels and their subunits within a broad eukaryotic context.”

Reviewer #3 (Comments to the Authors):

Peer Review | JGP Manuscript#: 202413636R1

Title: NALCN/Cch1 channelosome subunits originated in early eukaryotes and a complete set is conserved in animals, fungi, and apusomonads

The revised manuscript is much improved, easier to read, and the analysis and conclusions are more focussed.

MINOR COMMENTS

1. Page 6 of the PDF: '...predicted the structure of the human NALCN-FAM155A complex and structurally aligned it with its corresponding solved structure⁷, RECEALING highly overlapping structures with a root mean square alignment deviation score of 0.784 angstroms (Figure S3).! Correct the spelling mistake (REVEALING).

Thank you. We have made this correction (line 174).

1. Line 1 of the Discussion: 'Our phylogenetic analysis of four domain channels identified using a custom HMM profile trained on NALCN and Cch1 protein sequences...' I'm not sure that it is appropriate to say the the HMM profile was 'trained on' a given set of sequences. Perhaps a better way of saying it would be that 'it was built from'? In the age

of 'machine learning' the term 'trained on' means something very specific.

The you for pointing this out. We have changed this text to the following (lines 285 to 286):

“Our phylogenetic analysis of four domain channels identified using a custom HMM profile built from aligned NALCN and Cch1 protein sequences...”

Dr. Adriano Senatore
University of Toronto
Biology
3359 Mississauga Rd.
William G. Davis Building Rm. 3033
Mississauga, Ontario L5L 1C6
Canada

Re: 202413636R2

Dear Dr. Senatore,

I am pleased to let you know that your manuscript, entitled "NALCN/Cch1 channelosome subunits originated in early eukaryotes and a full complement is found in select animals, fungi, and apusomonads" is scientifically acceptable for publication in Journal of General Physiology. Formal acceptance will follow when it is modified in accordance with the referees' remarks and our editorial policies.

Please note items that need attention are listed at the bottom of this email (under 'manuscript formatting checklist') and on the attached marked-up pdf file. A shorter title would also be desirable, if possible. Please also be sure to include a copy of the text with alterations highlighted (boldfaced or underlined). Your manuscript should be a double-spaced MS Word file and include editable tables, if appropriate.

Lastly, JGP requires a data availability statement for all research article submissions. These statements will be published in the article directly above the Acknowledgments. The statement should address all data underlying the research presented in the manuscript. Please visit the JGP instructions for authors for guidelines and examples of statements at <https://rupress.org/jgp/pages/editorial-policies#data-availability-statement>.

Please submit your final files via this link:
Link Not Available

Thank you for choosing to publish your research in JGP and please feel free to contact me with any questions.

Sincerely,

Crina Nimigean, Ph.D.
On behalf of Journal of General Physiology

Journal of General Physiology's mission is to publish mechanistic and quantitative molecular and cellular physiology of the highest quality; to provide a best in class author experience; and to nurture future generations of independent researchers.

Manuscript formatting checklist:

- MS Word document of text needed (including editable tables)
- MS Word document of supplemental text needed, if applicable (including figure legends and editable tables)
- Brief Statement describing supplementary information needed, if applicable (in subsection at end of Materials & Methods)
- Please include a data availability statement preceding the Acknowledgments section. Please see <https://rupress.org/jgp/pages/editorial-policies#data-availability-statement>
- References need to follow JGP style (This article has numbered citations in it, which is not the style we use). Please refer to our guidelines here: <https://rupress.org/jgp/pages/reference-guidelines>
- Figures created at sufficient resolution and in acceptable format (including supplemental if applicable). If working in Illustrator, we prefer .ai or .eps file format. If working in Photoshop please use 600dpi/1000dpi .tiff or .psd file format. Minimum resolution at estimated print size: Minimum resolution for all figures is 600 dpi. For figures that contain both photographs and line art or text, 600 dpi is highly recommended. Figures containing only black and white elements (line art, no color, and no gray) should be 1,000 dpi. Maximum figure size is 7 in wide x 9 in high (17.5 x 22.8 cm) at the correct resolution. <https://jgp.rupress.org/fig-vid-guidelines>
- Supplemental figures, if any, conforming to same guidelines as manuscript figures (noted above)
- If images resemble one from a prior publications, the author must seek permissions (to reproduce or adapt) from the original publisher. [You can resubmit your paper while waiting to hear back from the original publisher but please keep us updated]
- All authors must complete a disclosure form prior to acceptance. A link to complete the form has been sent to all coauthors. Please provide the editorial office with updated email addresses if necessary